# Scheduling Thoughts: Learning the Order of Thought in Diffusion Language Models

**Jiawei Xu** [1]  **Minghui Liu** [1]  **Aakriti Agrawal** [1]  **Yifan Chen** [2][†]  **Furong Huang** [1][†]

## Abstract

Masked diffusion language models decode by iteratively unmasking tokens, where the unmasking order defines an "order of thought" that strongly influences generation quality yet is typically chosen heuristically. We derive a tractable upper bound on the sequential decoding mismatch, measured by the Kullback–Leibler divergence and expressed in terms of the model's pathwise log-likelihood, with tightness under sufficient model expressivity. This bound induces a dense self-aware reward over ordered trajectories, casting order selection as a principled policy optimization problem with a frozen denoiser. We instantiate this idea as **Self-Aware Scheduling (SAS)**, which learns a lightweight order policy using Group Relative Policy Optimization and applies seamlessly to both any-order and semi-autoregressive decoding. On Sudoku with 1B MDM, SAS improves puzzle accuracy from 82.0% (best heuristic schedule) to 91.8%, and reaches 97.5% with second-stage fine-tuning along learned trajectories. On mathematical reasoning with LLaDA-8B, SAS improves pass@1 on GSM8K from 64% to 76% and on MBPP from 39.5% to 41%, consistently matching or exceeding heuristic schedules across generation lengths and block sizes. Project page: https://jimmyxu123.github.io/SAS.

## 1. Introduction

**Diffusion decoding has a hidden degree of freedom: *the order of thought*.** Masked diffusion language models (Austin et al., 2021; Lou et al., 2023; Hoogeboom et al., 2021b; Shi et al., 2024) generate discrete sequences by iteratively unmasking tokens, offering a flexible alternative to the fixed left-to-right factorization of autoregressive language models (Radford et al., 2018). Crucially, diffusion decoding is not tied to a single generation order: inference depends on an *unmasking schedule* that decides which positions are revealed at each step. This schedule is more than an implementation detail: it is an implicit "order of thought", and it determines which commitments the model makes early, which constraints it propagates, and which ambiguities it strategically postpones. Because different schedules trace different conditional trajectories, they induce different output distributions, and the schedule choice can substantially affect generation quality (Kim et al., 2025; Zheng et al., 2023).

**Heuristic schedules are fast, but they can think myopically.** Most existing schedules are fixed greedy heuristics based on per-position uncertainty (e.g., confidence, margin, entropy) (Zheng et al., 2023; Kim et al., 2025; Ben-Hamu et al., 2025). While efficient, such rules optimize the *next* step rather than the *whole* decoding trajectory: they often pick "easy" tokens that become locally certain under the current partial context, even when a different reveal would better shape future conditionals. Empirically, we find that these myopic choices underperform on complex reasoning tasks (see Table 1 and Figure 2 in Section 6.1). Even worse, expert-designed logical orders can be suboptimal for an *imperfect* pretrained diffusion model: what is logically natural for humans is not necessarily the path along which the model's conditional predictions are most reliable (see Section 6.1). These observations motivate a principled approach that *learns* the order, instead of hard-coding it.

**Key idea and theory: learn the schedule by optimizing the model's *trajectory likelihood*.** We treat diffusion decoding as a latent-order generative process (Huang et al., 2025; Wang et al., 2025d) and ask a direct question: *which reveal order makes the model most faithful to the data distribution?* Concretely, we formalize scheduling as minimizing the KL divergence between the data distribution and the distribution induced by a given decoding procedure. Our main theoretical contribution is to show that this intractable objective admits a tractable bound expressed in terms of the denoiser's *pathwise* (trajectory) log-likelihood under teacher forcing. This yields a dense, model-self-aware reward that assigns credit to *each* reveal decision, rather than only the

---

[†]Equal advising. [1]University of Maryland, College Park [2]University of California, Los Angeles. Correspondence to: Yifan Chen <yifanchen@math.ucla.edu>, Furong Huang <furongh@umd.edu>.

*Proceedings of the 43$^{rd}$ International Conference on Machine Learning*, Seoul, South Korea. PMLR 306, 2026. Copyright 2026 by the author(s).

final sample. Under this reward, a good schedule reveals the token maximizing the expected *future* log-likelihood of the remaining trajectory, i.e., it chooses the next "thought" to make downstream reasoning easiest for the model, aligning the decoding path with the model's predictive strengths rather than with local uncertainty alone.

**Method: Self-Aware Scheduling (SAS) for plug-and-play order learning.** Guided by the theory, we propose **Self-Aware Scheduling (SAS)**, a lightweight framework that learns an order policy while keeping the diffusion model fixed. At each decoding step, the policy outputs a distribution over currently masked positions and selects (or samples) one position to reveal next. SAS is plug-and-play (no retraining of the base denoiser), and it can be optimized efficiently with Group Relative Policy Optimization (GRPO) (Shao et al., 2024). The same learned policy applies seamlessly to both any-order sequential decoding and semi-autoregressive (block) decoding (Arriola et al., 2025), turning "how to unmask" into a trainable component of inference.

**Results: optimizing the order of thought is a strong lever for reasoning.** Across diverse reasoning benchmarks, SAS yields consistent and substantial improvements over heuristic and expert-designed schedules. On large-scale Sudoku with a 1B masked diffusion model, SAS improves puzzle accuracy from $82.0\%$ (best heuristic) to $91.8\%$. On mathematical reasoning (GSM8K) with LLaDA-8B (Nie et al., 2025b), it improves pass@1 from $64\%$ to $76\%$. We further show that a simple second-stage fine-tuning procedure along the learned trajectories using our self-aware objective provides additional gains. Together, these results position decoding order as a controllable and optimizable reasoning primitive: beyond learning *what* to generate, we can learn *when* to commit to each piece of information—in effect, learning an order of thinking that the model itself finds most reliable.

**Summary of Contributions.**

- **Theory (self-aware objective for order learning):** We cast diffusion decoding as a latent-order generative process and derive a tractable bound linking decoding mismatch (KL divergence) to the denoiser's cumulative pathwise log-likelihood, yielding a dense, model-self-aware reward for credit assignment over reveal decisions.

- **Algorithm (SAS):** We introduce **Self-Aware Scheduling**, a plug-and-play framework that learns an unmasking-order policy for a *frozen* diffusion LM via GRPO, applicable to both any-order and semi-autoregressive decoding.

- **Empirical validation (reasoning gains + second stage):** We demonstrate large, consistent improvements over heuristic and expert schedules on Sudoku,

math and coding reasoning, and show additional gains from second-stage fine-tuning along learned trajectories under the same self-aware objective.

## 2. Preliminaries and Problem Setup

### 2.1. Masked Diffusion Models

Masked diffusion models generate discrete sequences by iteratively demasking tokens, analogous to the denoising process in continuous diffusion models (Song et al., 2020; Ho et al., 2020). Let $\mathcal{X}$ denote a finite vocabulary and $n$ the sequence length. A sequence is $x = (x_1, \ldots, x_n) \in \mathcal{X}^n$. For a mask pattern $M \subseteq [n] := \{1, \ldots, n\}$, define the masked sequence $x^M \in (\mathcal{X} \cup \{\,\texttt{[MASK]}\,\})^n$ by

$$(x^M)_i := \begin{cases} \texttt{[MASK]} & \text{if } i \in M, \\ x_i & \text{if } i \notin M. \end{cases} \tag{1}$$

A masked diffusion model (or *denoiser*) parameterized by $\theta$ defines conditional distributions over masked tokens given observed context. Specifically, **the model** $p_\theta$ provides position-wise predictions $p_\theta^i(\cdot \mid x^M)$, $i \in M$, representing the distribution over token $x_i$ conditioned on the partial observation $x^M$. Here $\theta$ denotes the learned parameters of the base model.

### 2.2. Unmasking Schedules: the Order of Thought

Generation proceeds by iteratively unmasking positions, a process known as *decoding*. The unmasking schedule can be interpreted as an "order of thought" for generation. The flexibility of the masked diffusion model is that it allows any-order decoding in principle, and thus the unmasking order, or schedule, matters. The **goal** of our paper is to introduce a methodology to learn the unmasking order. We define the schedule $v_\phi$, parameterized by $\phi$, as the "**order policy**", which takes a partial sequence as input and outputs a distribution over which position(s) to unmask next.

We denote by $\tilde{x}^{(t)}$ the partial sequence at decoding step $t$, where some positions contain tokens from $\mathcal{X}$ (revealed) and others contain $\texttt{[MASK]}$ (not yet revealed). Starting from the fully masked sequence, at each step the algorithm selects which position(s) to unmask and samples their values from the model. We distinguish between two decoding regimes: *sequential decoding*, which unmasks a single position per step, and *parallel decoding*, which unmasks multiple positions simultaneously. In this work, we focus on the sequential setting to isolate the effects of ordering, while the parallel regime is detailed in Appendix A.1.

**Unmasking procedure.** Let $S_t \subseteq [n]$ denote the set of revealed positions at step $t$, and $M_t = [n] \backslash S_t$ denote the set of masked positions. Starting from $\tilde{x}^{(0)} = x^{[n]}$ (fully masked) with $S_0 = \emptyset$, at each step $t = 1, \ldots, n$: **(1)** Select

next position: $i_t \sim v_\phi(\cdot \mid \tilde{x}^{(t-1)})$ where $v_\phi(\cdot \mid \tilde{x}^{(t-1)})$ outputs a distribution over $[n] \setminus S_{t-1}$. **(2)** Sample token: $\tilde{x}_{i_t}^{(t)} \sim p_\theta^{i_t}(\cdot \mid \tilde{x}^{(t-1)})$. **(3)** Update: $S_t \leftarrow S_{t-1} \cup \{i_t\}$; set $\tilde{x}_j^{(t)} = \tilde{x}_j^{(t-1)}$ for all $j \neq i_t$. The complete run produces an *unmasking order* $\sigma = (i_1, \ldots, i_n)$ and final sequence $\tilde{x}^{(n)} \in \mathcal{X}^n$.

**Induced distributions.** The decoding procedure induces distributions over output sequences. Characterizing these distributions is essential to quantify the discrepancy between the generative process and the true data distribution, thereby providing a principled objective for optimizing the order policy. To achieve so, we define notations for the decoding algorithm under *teacher forcing* (Williams & Zipser, 1989), where the policy $v_\phi$ and model $p_\theta$ are evaluated on partially revealed versions of a fixed target sequence $x$, as will be explained below. This allows us to compute the probability the algorithm assigns to generating $x$ under a particular unmasking order.

For a target sequence $x$ and unmasking order $\sigma = (i_1, \ldots, i_n)$, define the *teacher-forced trajectory* $\tilde{x}^{(0)}(\sigma), \ldots, \tilde{x}^{(n)}(\sigma)$ where $\tilde{x}^{(t)}(\sigma)$ reveals positions $\{i_1, \ldots, i_t\}$ from $x$:

$$\tilde{x}^{(t)}(\sigma)_j := \begin{cases} x_j & \text{if } j \in \{i_1, \ldots, i_t\}, \\ [\texttt{MASK}] & \text{otherwise.} \end{cases} \quad (2)$$

In other words, $\tilde{x}^{(t)}(\sigma)$ is the partial sequence obtained by revealing the first $t$ positions in order $\sigma$ using their values from target $x$. Under teacher-forcing with target $x$ and order $\sigma$, the *order policy* (parameterized by $\phi$) assigns probability

$$v_\phi(\sigma \mid x) := \prod_{t=1}^n v_\phi(i_t \mid \tilde{x}^{(t-1)}(\sigma)) \quad (3)$$

to the unmasking order $\sigma$, where each factor evaluates the policy $v_\phi$ on the teacher-forced state $\tilde{x}^{(t-1)}(\sigma)$. The *model* (parameterized by $\theta$) assigns path likelihood

$$p_\theta(x \mid \sigma) := \prod_{t=1}^n p_\theta^{i_t}(x_{i_t} \mid \tilde{x}^{(t-1)}(\sigma)), \quad (4)$$

where each factor evaluates the model $p_\theta$ at position $i_t$ given teacher-forced state $\tilde{x}^{(t-1)}(\sigma)$. The sequential algorithm induces the joint distribution over sequences and orders

$$P_{\theta,\phi}^{\text{seq}}(x, \sigma) := v_\phi(\sigma \mid x) \cdot p_\theta(x \mid \sigma), \quad (5)$$

with marginal distribution over outputs

$$P_{\theta,\phi}^{\text{seq}}(x) = \sum_\sigma v_\phi(\sigma \mid x) \cdot p_\theta(x \mid \sigma). \quad (6)$$

*Remark* 2.1. Heuristic schedules can be formalized as a special case of our order policy framework where the policy $v_\phi$ is deterministic and non-trainable. Details are provided in Appendix A.3.

## 3. Unmasking Order Optimization: From Error Bounds to Reward Design

*Figure 1.* High-level overview of Section 3: We overcome the intractability of latent ordering by deriving a theoretical framework that allows us to optimize a tractable, self-aware reward.

This section formalizes how the unmasking order affects the induced decoding distribution. Our primary goal is to derive a principled objective for optimizing the unmasking policy in *sequential* decoding. We also characterize the additional discrepancy introduced by *parallel* decoding via an information-theoretic quantity (Section B.1).

Let $\pi(x)$ be the target distribution on $\mathcal{X}^N$ and let $P_{\theta,\phi}^{\text{seq}}(x)$ be the output law induced by *sequential* decoding (Section 2.1). We aim to fit the induced decoder distribution to the data by maximizing its expected log-likelihood,

$$\max_{\theta,\phi} \mathbb{E}_{x \sim \pi} \big[ \log P_{\theta,\phi}^{\text{seq}}(x) \big], \quad (7)$$

which is equivalent to minimizing $\text{KL}(\pi \| P_{\theta,\phi}^{\text{seq}})$ up to the constant $H(\pi)$.

### 3.1. Self-aware Reward as Pathwise Likelihood

For a data sample $x$ and an unmasking order $\sigma = (i_1, \ldots, i_n)$, let $p_\theta(x \mid \sigma)$ denote the pathwise likelihood assigned by the diffusion model when the tokens of $x$ are revealed according to $\sigma$ under teacher forcing. We define the *self-aware reward* as

$$R_\theta(x, \sigma) := \log p_\theta(x \mid \sigma). \quad (8)$$

Thus, the reward is simply the frozen model's own pathwise log-likelihood evaluated along a specific order. It scores an order by how well the diffusion model can explain the data when forced to reveal tokens in that order.

We train the order policy by minimizing the negative expected self-aware reward:

$$\min_{\theta,\phi} \mathcal{L}_{\text{SAS}}(\theta, \phi) := \min_{\theta,\phi} -\mathbb{E}_{x \sim \pi} \mathbb{E}_{\sigma \sim v_\phi(\cdot \mid x)} \big[ R_\theta(x, \sigma) \big]. \quad (9)$$

In this work we primarily learn $\phi$ with $\theta$ frozen, and optionally apply a second-stage fine-tuning of $\theta$ under a fixed learned order (Section 6.4).

We next record two simple facts connecting this pathwise objective to the marginal likelihood over orders. These results are not needed to motivate the reward itself; rather, they clarify how the reward relates to the likelihood-based objective.

**Theorem 3.1** (Pathwise likelihood lower bound). *For any $\pi$ and order policy $v_\phi$, the marginal log-likelihood satisfies*

$$\mathbb{E}_{x \sim \pi}\big[\log P_{\theta,\phi}^{\text{seq}}(x)\big] \geq \mathbb{E}_{x \sim \pi}\,\mathbb{E}_{\sigma \sim v_\phi(\cdot|x)}\big[\log p_\theta(x \mid \sigma)\big]. \tag{10}$$

*Proof.* Since $P_{\theta,\phi}^{\text{seq}}(x) = \mathbb{E}_{\sigma \sim v_\phi(\cdot|x)}[p_\theta(x \mid \sigma)]$, Jensen's inequality gives $\log P_{\theta,\phi}^{\text{seq}}(x) \geq \mathbb{E}_{\sigma \sim v_\phi(\cdot|x)}[\log p_\theta(x \mid \sigma)]$. Taking expectation over $x \sim \pi$ proves the claim. □

### 3.2. KL Interpretation for Sequential Decoding

We also relate the self-aware loss to the marginal mismatch $\text{KL}(\pi \| P_{\theta,\phi}^{\text{seq}})$. This gives a compact interpretation of the pathwise objective as a joint distribution matching objective over data and orders.

**Theorem 3.2** (Joint KL identity and marginal bound). *Let $Q_\phi(x,\sigma) := \pi(x)v_\phi(\sigma|x)$ and $P_{\theta,\phi}(x,\sigma) := p_\theta(x|\sigma)v_\phi(\sigma|x)$ be the data-policy and model-policy joint distributions, respectively. The self-aware loss $\mathcal{L}_{\text{SAS}}$ satisfies*

$$\begin{aligned}\text{KL}(\pi(x)\|P_{\theta,\phi}^{\text{seq}}(x)) &\leq \mathcal{L}_{\text{SAS}}(\theta,\phi) - H(\pi) \\ &= \text{KL}(Q_\phi(x,\sigma)\|P_{\theta,\phi}(x,\sigma)).\end{aligned} \tag{11}$$

*where $H(\pi)$ is the constant data entropy.*

The proof is provided in Appendix B.2. Theorem 3.2 shows that minimizing the self-aware loss is equivalent, up to the constant $H(\pi)$, to minimizing a joint KL mismatch over $(x, \sigma)$. This joint mismatch also upper bounds the marginal generation mismatch $\text{KL}(\pi(x)\|P_{\theta,\phi}^{\text{seq}}(x))$.

**Corollary 3.3** (Joint realizability and tightness). *If the model family is expressive enough to allow for joint realizability—i.e., there exist parameters $(\theta^\star, \phi^\star)$ such that $\text{KL}(Q_{\phi^\star}(x,\sigma)\|P_{\theta^\star,\phi^\star}(x,\sigma)) = 0$—then the marginal mismatch $\text{KL}(\pi\|P_{\theta,\phi}^{\text{seq}})$ is zero, and the pathwise lower bound of Theorem 3.1 is tight, satisfying $\mathcal{L}_{SAS}(\theta^\star, \phi^\star) = H(\pi)$.*

The proof is provided in Appendix B.3. Corollary 3.3 states that under joint realizability the bound is achievable, yielding $\text{KL}(\pi(x)\|P_{\theta,\phi}^{\text{seq}}(x)) = 0$. More generally, if approximate realizability holds with $\text{KL}(Q_{\phi^\star}(x,\sigma)\|P_{\theta^\star,\phi^\star}(x,\sigma)) \leq \epsilon$, then the marginal mismatch is also bounded as $\text{KL}(\pi(x)\|P_{\theta,\phi}^{\text{seq}}(x)) \leq \epsilon$.

*Remark* 3.4. In the regime of a perfect diffusion model $p_\theta$, the upper bound in Theorem 3.2 is tight with $\mathcal{L}_{\text{SAS}}(\theta^\star, \phi) = H(\pi)$ for *any* order policy $\phi$. Thus, order learning is most useful when the denoiser is imperfect: improving $\phi$ finds easier generation paths for the frozen denoiser, while improving $\theta$ improves the pathwise predictions. Both reduce the joint mismatch, as we observe empirically in Section 6.4. The proof is in Appendix B.4.

*Remark* 3.5. Although the main text focuses on sequential decoding, the same self-aware loss also provides control over parallel decoding mismatch. In Appendix B, we show that the additional error caused by block updates is upper-bounded by a term governed by within-block conditional dependencies, which, together with the self-aware loss, upper bounds the total variation error of the generated distribution.

## 4. Methodology and Policy Optimization

We focus on *sequential* (one-by-one) unmasking, in which the decoding procedure reveals exactly one position per step. Concretely, we freeze the diffusion model parameters $\theta$ (and its denoising head) and optimize only the order-policy parameters $\phi$ using reinforcement learning.

### 4.1. Learning with Self-aware Reward

**Monte Carlo approximation of the objective.** For a target sequence $x \sim \pi$ and order $\sigma$, we define the return as the **pathwise log-likelihood** of the target tokens under the frozen denoiser:

$$R_\theta(x, \sigma) = \sum_{t=1}^{N} \log p_\theta^{i_t}(x_{i_t} \mid \tilde{x}^{(t-1)}(\sigma)). \tag{12}$$

This serves as a dense, self-aware signal for credit assignment.

**Order-policy objective (optimize $\phi$ only).** We learn the order policy $v_\phi(\sigma \mid x)$ by maximizing this expected return while keeping the diffusion model $\theta$ fixed:

$$\phi^\star \in \arg\max_\phi \mathbb{E}_{x \sim \pi}\,\mathbb{E}_{\sigma \sim v_\phi(\cdot|x)}\Big[R_\theta(x, \sigma)\Big]. \tag{13}$$

### 4.2. Parameterization of the Order Policy

**State-dependent categorical policy over remaining indices.** At step $t$, the policy observes the current partial state $s_{t-1} := \tilde{x}^{(t-1)}$ and chooses the next index $i_t$ from the masked set $M_{t-1}$. We parameterize the order policy as a categorical distribution over indices:

$$\begin{aligned}v_\phi(\sigma \mid x) &= \prod_{t=1}^{N} v_\phi(i_t \mid s_{t-1}), \\ v_\phi(i \mid s_t) &= 0 \text{ if } i \notin M_t.\end{aligned} \tag{14}$$

**Scoring form (masked softmax).** Given a partial state $s_t$, we compute a scalar score $u_{\phi,i}(s_t) \in \mathbb{R}$ for each position

$i \in [N]$ and define

$$v_\phi(i \mid s_t) \;=\; \frac{\exp\big(u_{\phi,i}(s_t)/\tau\big)\,\mathbf{1}\{i \in M_t\}}{\sum_{j \in M_t} \exp\big(u_{\phi,j}(s_t)/\tau\big)}, \qquad (15)$$

where $\tau > 0$ is a temperature. In our implementation, $u_{\phi,i}(s_t)$ is produced by a lightweight policy head on top of frozen diffusion-model features. We refer the reader to Appendix C for details of order policy design.

### 4.3. GRPO training

**MDP formulation.** Order learning can be cast as a finite-horizon Markov Decision Process (MDP):

- **State** $s_{t-1}$: the partially revealed sequence $\tilde{x}^{(t-1)}$
- **Action** $a_t \in M_{t-1}$: choose the next index $i_t$ to unmask.
- **Transition**: deterministic teacher-forced reveal, $s_t = \mathcal{T}(s_{t-1}, a_t)$ obtained by setting position $i_t$ to $x_{i_t}$.
- **Reward**: $r_t = \log p_\theta^{i_t}(x_{i_t} \mid s_{t-1})$.

The trajectory corresponds to an order $\sigma$, and the return equals the self-aware reward $R_\theta(x, \sigma)$.

**Group Relative Policy Optimization (GRPO).** We optimize $v_\phi$ using GRPO (Shao et al., 2024), a critic-free method that employs group-relative normalization. For each input $x$, we sample a group of $G$ outputs $\{\sigma^{(i)}\}_{i=1}^G$ from the old policy $v_{\phi_{\text{old}}}$. An advantage $\widehat{A}_i$ is computed by standardizing the cumulative reward $R_i$ against the group statistics: $\widehat{A}_i = (R_i - \mu_G)/(\sigma_G + \epsilon_{\text{adv}})$. The GRPO objective maximizes a clipped surrogate function:

$$J_{\text{GRPO}}(\phi) = \mathbb{E}\Bigg[\frac{1}{G}\sum_{i=1}^G \frac{1}{N}\sum_{t=1}^N \min\Big(g_{i,t}(\phi)\hat{A}_{i,t},$$
$$\text{clip}\big(g_{i,t}(\phi), 1-\epsilon, 1+\epsilon\big)\hat{A}_{i,t}\Big)\Bigg] \qquad (16)$$

with the per-step probability ratio $g_{i,t} = \frac{v_\phi(\sigma_t^{(i)}|s_{t-1}^{(i)})}{v_{\phi_{\text{old}}}(\sigma_t^{(i)}|s_{t-1}^{(i)})}$.

Each training example pairs a prompt $c$ with a target completion $x$. The policy operates exclusively on the completion positions, treating the prompt as fixed context. Thus, the process begins at $s_0 = \text{concat}(c, \tilde{x}^{(0)})$, where the prompt is fully visible and the target is entirely masked.

## 5. Related Work

**Heuristic schedules for diffusion LLMs.** The unmasking schedule critically determines information accrual in masked diffusion models (Nie et al., 2025b; Ye et al., 2025; Shi et al., 2024). Prior work predominantly uses training-free heuristics that prioritize tokens by confidence or local certainty (Ben-Hamu et al., 2025; Wei et al., 2025; Hong

et al., 2025; Kim et al., 2025; Li et al., 2025; Yu et al., 2025), with variants incorporating spatial-temporal structure (Huang et al., 2026; Wang et al., 2025b) or token dependencies (Azangulov et al., 2025). We instead learn a sequential policy via RL to optimize for global trajectory coherence.

**Learning orders.** Finding effective decoding orders has been explored in non-autoregressive generation (Wu et al., 2018; Zhu et al., 2019; Gu et al., 2019; Welleck et al., 2019; Stern et al., 2019). For diffusion LMs, recent work optimizes schedules using task-specific verifiable rewards: Huang et al. (2025); Zhao et al. (2025a) jointly train the policy and MDM, while Hong et al. (2025); Jazbec et al. (2025) learn policies for frozen models using sparse terminal rewards. We propose minimizing KL divergence—a general objective that yields dense rewards and applies even when verifiable rewards are unavailable. Since discrete diffusion equivalently represents any-order autoregressive models (Hoogeboom et al., 2021a; Ou et al., 2024), related approaches derive similar ELBO-based objectives for diffusion molecular generation (Wang et al., 2025d), though requiring two additional networks and more complex formulation. We also note related work on optimizing orders for efficient parallel decoding to accelerate generation (Chen et al., 2024; Shih et al., 2023; Park et al., 2024), complementary to the statistical efficiency we pursue. Our theoretical bounds extend to parallel decoding (Appendix B.1).

**Diffusion LLMs post-training.** Recent RL-based post-training focuses on refining the diffusion backbone itself. For instance, D1 (Zhao et al., 2025b) introduces a GRPO variant tailored for discrete diffusion, while other works concentrate on improving policy gradient estimation (Tang et al., 2025; Wang et al., 2025c; Lin et al., 2025; Zhu et al., 2025; Wang et al., 2025a) to enhance reasoning capabilities. These approaches typically use fixed confidence-based schedules during optimization. In contrast, we propose a two-stage framework prioritizing generation order: first optimizing the unmasking policy with a frozen diffusion head, then fine-tuning the head along learned trajectories. This approach is computationally efficient without expensive RL updates on the full model.

## 6. Experiments

In this section, we empirically validate that our self-aware reward formulation is both effective and scalable. Our experiments span diverse reasoning benchmarks—including logic puzzles, mathematical reasoning, and code generation—and cover model scales ranging from 1B to 8B parameters.

**Experimental Objectives.** Our evaluation is guided by three primary research questions:(i) **Performance:** Can optimizing the unmasking order alone yield consistent improvements in reasoning tasks? (ii) **Emergent Strategy:**

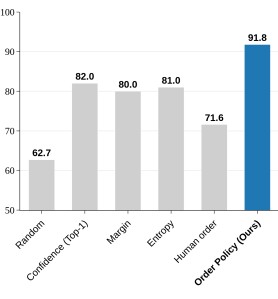

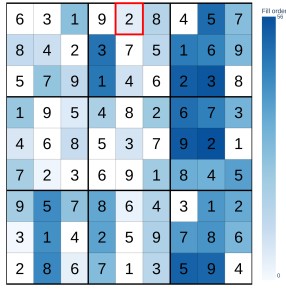

*(a)* Accuracy         *(b)* Order Visualization

*Figure 2.* **Left:** Our method (Order Policy) achieves significantly higher success accuracy (91.8%) on Sudoku puzzles compared to all baselines. **Right:** Visualization of the learned unmasking order. White cells represent fixed initial hints. Blue cells indicate generated tokens, where color intensity corresponds to the decoding step: lighter blue denotes early unmasking (high-confidence/foundational moves), while darker blue denotes later unmasking. For example, we red-boxed the first move located, which corresponds to the *naked single* of the initial state—the easiest position to solve.

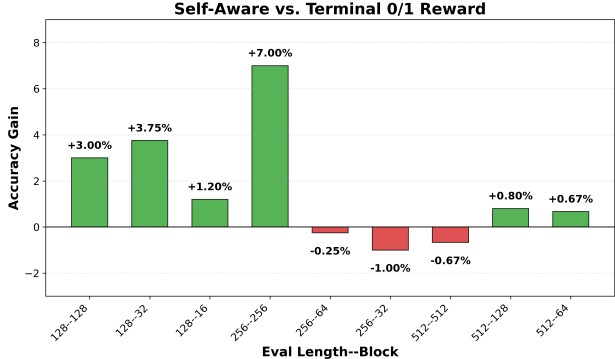

*Figure 3.* Self-aware reward vs. $0/1$ terminal reward for learning the order policy on GSM8K (5-shot). Notation `L--B`: total length $L$ and block size $B$. Bars represent the accuracy difference in percentage points.

What decoding strategies emerge from the learned policy, and how do they correlate with the underlying task structure? (iii) **Generalization:** Does the learned order policy generalize across varying sequence lengths and decoding modalities (e.g., full diffusion vs. semi-autoregression)?

We investigate (i) across all benchmarks, restrict our structural analysis (ii) to the interpretable domain of Sudoku, and assess (iii) using mathematical and code generation tasks.

### 6.1. Sudoku: Controlled Study of Unmasking Order

**Setup.** We construct a large-scale Sudoku corpus from (Rao, 2019). From the provided training split, we subsample 1M puzzles for training. Each Sudoku board is represented as a length-81 discrete sequence where each token corresponds to one cell value and $0$ denotes the blank cell. We treat $0$

*Table 1.* Sudoku performance under different unmasking-order strategies. We report *puzzle accuracy* (fraction of boards solved) and *cell accuracy* (fraction of correctly filled cells).

| ORDER STRATEGY | PUZZLE ACC. (%) | CELL ACC. (%) |
|---|---|---|
| RANDOM | 62.7 | 89.80 |
| CONFIDENCE (TOP-1) | 82.0 | 98.56 |
| MARGIN | 80.0 | 98.38 |
| ENTROPY | 81.0 | 98.53 |
| HUMAN ORDER | 76.6 | 96.18 |
| ORDER POLICY (OURS) | **91.8** | **99.21** |

as the `[MASK]` token and model the completion of masked cells as conditional generation.

We instantiate with a 1B-scale masked diffusion model: SMDM-1B (Nie et al., 2025a). We first fine-tune the diffusion model on the 1M training puzzles for one epoch; we then train a lightweight MLP policy on 1,000 puzzles (sampled from the training dataset) and report results on the remaining test split.

**Baselines.** At inference time, we compare the learned policy against heuristic decoding orders: (i) RANDOM (Zheng et al., 2024); (ii) CONFIDENCE (Zheng et al., 2023); (iii) MARGIN (Kim et al., 2025); (iv) ENTROPY (Zheng et al., 2023); and (v) a human EXPERT order (Shah et al., 2024) based on Sudoku solving logic.

**Results.** Table 1 reports puzzle and cell accuracy across all order baselines. Our learned order policy achieves the best performance by a large margin, confirming that **optimizing unmasking order alone can substantially improve structured reasoning**. Interestingly, the human EXPERT order underperforms: it is approximately $5\%$ worse than CONFIDENCE and about $15\%$ worse than our learned policy in puzzle accuracy. This motivates a deeper analysis of the order strategies in Sudoku. To understand the decision logic of our Self-Aware Schedule, we visualize the decoding order of a specific puzzle in Figure 2b. The resulting heatmap demonstrates a clear "easy-to-hard" curriculum. The policy prioritizes deterministic unmasking cells given the current context (e.g., naked singles). By resolving these high-certainty tokens first (light blue), the model propagates constraints to the more ambiguous regions (dark blue).

### 6.2. Order Analysis of Sudoku and Kendall's $\tau$

To analyze how different schedules align with human expert ordering—while accounting for the fact that many Sudoku moves are interchangeable—we introduce an equivalence-class variant of Kendall's $\tau$ (Kendall, 1948) which is agnostic to the relative ordering of moves that are strategy-equivalent. We provide details in Appendix D.1.

**Results.** Table 2 reports the mean $\tau_{\text{eq}}$, the fraction of puzzles with $\tau_{\text{eq}} > 0$, and the fraction with $\tau_{\text{eq}} > 0.5$. While

*Table 2.* Comparison with human expert order on Sudoku via equivalence-class Kendall's $\tau$ (2000 puzzles).

| ORDER STRATEGY | MEAN $\tau_{eq}$ | FRAC. $\tau_{eq} > 0$ | FRAC. $\tau_{eq} > 0.5$ |
|---|---|---|---|
| RANDOM | -0.002 | 0.49 | 0.00 |
| CONFIDENCE (TOP-1) | 0.514 | 1.00 | 0.58 |
| MARGIN | 0.516 | 0.99 | 0.59 |
| ENTROPY | 0.513 | 1.00 | 0.58 |
| **ORDER POLICY (OURS)** | **0.735** | **0.59** | **0.47** |

our learned order policy attains the highest agreement with expert ordering, the alignment remains imperfect ($\tau_{eq} < 1$), suggesting that the model's optimal path differs from human logical progression. When viewed alongside the performance gains in Table 1, this observation suggests a critical insight: for masked diffusion LMs, strict adherence to human solving orders is **not** a prerequisite for optimal performance, and the model may benefit from alternative reasoning pathways. In other words, expert logic provides a strong prior over precedence constraints, but the optimal schedule for a pretrained diffusion decoder is shaped from modeling uncertainty and statistical correlations, which our self-aware objective explicitly exploits.

### 6.3. Generalization to Math and Code Reasoning

**Setup.** We then move to study whether an order policy learned with the self-aware objective transfers to larger-scale reasoning tasks beyond Sudoku. We use LLADA-8B-INSTRUCT (Nie et al., 2025b) as the frozen base masked diffusion model and parameterize the order policy as a lightweight one-layer Transformer. We train the policy under the full diffusion (any-order) sequential decoding paradigm from the training splits of GSM8K (Cobbe et al., 2021) and MBPP (Austin et al., 2021). The policy is trained with teacher-forced trajectories at a maximum generation length of 512. Training details and inference costs are provided in Appendix D.2.

**Evaluation protocol.** We evaluate (i) generalization across generation lengths and (ii) transfer to semi-autoregressive (block) inference. For length generalization, we report results for target lengths in $\{128, 256, 512\}$. For semi-autoregressive decoding, we vary the number of decoding blocks $K \in \{1, 4, 8\}$ and apply the same learned policy to schedule these blocks at inference time. We denote each setting as `promptlen/total-block`, e.g., `256--64` indicates total length 256 with block size 64 (semi-autoregressive), while `256--256` corresponds to full diffusion decoding. We evaluate the results using `lm-eval-harness`.

**Results.** In Figure 4, it demonstrates the robust generalization of our approach on GSM8K and MBPP. While distinct heuristic baselines exhibit inconsistent performance across different settings, our learned order policy uniquely

*Table 3.* Second-stage fine-tuning of the diffusion model with a fixed unmasking order on Sudoku. Fine-tuning yields significant gains across schedules: Human Order: Accuracy improves from $76.6\% \to 97.8\%$. Order Policy (Ours): Accuracy improves from $91.8\% \to 97.5\%$. Notably, our self-discovered curriculum achieves parity with human expertise without needing ground-truth traces.

| FIXED ORDER FOR FINE-TUNING | BEFORE | AFTER |
|---|---|---|
| HUMAN ORDER | 76.6 | 97.8 |
| ORDER POLICY (OURS) | 91.8 | 97.5 |

achieves superior or competitive performance across the entire spectrum of decoding regimes. Notably, we show while Left-to-Right scheduled decoding is competitive at shorter lengths ($L = 128$), our order policy achieves superior performance even in full diffusion generation at longer lengths. This confirms that our scheduling strategy captures fundamental reasoning structures and is highly robust to shifts in generation horizon, effectively transferring well beyond its training configuration.

### 6.4. Second-Stage Fine-Tuning with the Self-Aware Objective

Corollary 3.3 shows that, in the realizable setting, jointly optimizing the unmasking order policy and the diffusion model can in principle drive the sequential decoding error to zero. Motivated by this, we propose a simple *second-stage* optimization: after learning an order policy, we freeze the policy and further fine-tune the diffusion model using the same self-aware objective.

**Fixed trajectory optimization.** The self-aware objective (Eq. (9)) can be used to optimize both $\theta$ and $\phi$. After learning an order policy $\bar{v}$, we fix it and replace the stochastic order distribution by a deterministic rollout determined by $\bar{v}$. This reduces the training signal from a *corpus of possible orders* to a *single order trajectory*, yielding a simple supervised fine-tuning loss for the diffusion model.

**Second-stage objective.** With the fixed order $\hat{\sigma}$, the optimization over $\theta$ becomes

$$\max_{\theta} \mathbb{E}_{x \sim \pi}\Big[ R_{\theta}\big(x, \hat{\sigma}\big)\Big] \equiv \min_{\theta} \mathcal{L}_{\text{SFT}}(\theta), \quad (17)$$

which is the negative log-likelihood of ground-truth tokens *along the fixed unmasking schedule*.

We validate this second-stage optimization on Sudoku using two fixed orders: (i) the learned order policy $\bar{v}$ (greedy rollout), and (ii) a human expert order. Table 3 reports puzzle accuracy before and after second-stage fine-tuning. With the learned order, fine-tuning the diffusion head substantially improves puzzle accuracy, from 91.8% to 97.5%. We also observe that fine-tuning under the human expert order leads to a large improvement from 76.6% to 97.8%.

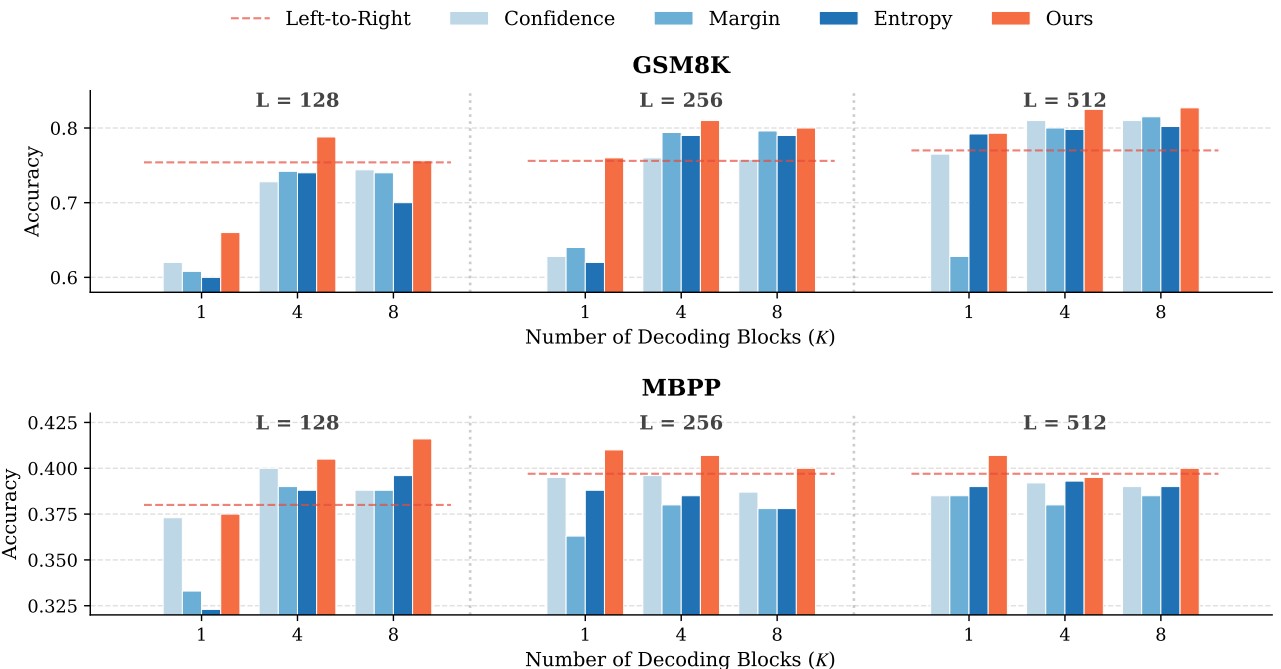

*Figure 4.* **Comparison across generation length and semi-autoregressive decoding.** We evaluate our proposed order policy against the other baselines across varying sequence lengths ($L$) and block numbers. The red dashed line represents standard Left-to-Right decoding heuristic of LLaDA-8B. Our method consistently outperforms the baseline. Notably, while Left-to-Right decoding is competitive at shorter lengths ($L = 128$), our order policy still achieves superior performance even in full diffusion generation at longer lengths.

Notably, our order learning matches the performance of supervised learning under expert ordering, demonstrating that the learned policy discovers a curriculum as effective as human expertise without requiring ground-truth ordering traces. Ultimately, these results underscore the **importance of scheduling thoughts effectively before refining the thoughts themselves**. While this work primarily focuses on learning the order schedule, it demonstrates that our self-aware objective serves as an effective and efficient post-training method on diffusion model weights, achieving significant gains without complicated RL on the model.

### 6.5. Cross-domain Transfer of Learned Order Policies

To evaluate whether the learned order policy captures reusable scheduling behavior rather than only task-specific patterns, we test cross-domain transfer between math and code reasoning tasks. In particular, we train the order policy on one domain and evaluate it on the other without further task-specific adaptation. Table 4 reports the transferred policy's performance under different decoding settings, where $L$-$B$ denotes generation length and block size. The value in parentheses is the gain over the strongest heuristic baseline under the same setting.

The transferred policy outperforms the best heuristic baseline in all tested settings. This suggests that SAS does not

*Table 4.* Cross-domain transfer of learned order policies. "Math $\rightarrow$ Code" denotes training the order policy on math and evaluating it on code, while "Code $\rightarrow$ Math" denotes the reverse. Numbers in parentheses indicate gains over the strongest heuristic baseline under the same decoding setting.

| Transfer setting | 256-256 | 256-64 | 256-32 |
|---|---|---|---|
| Math $\rightarrow$ Code | 0.41 (+1.5%) | 0.40 (+1.4%) | 0.39 (+0.33%) |
| Code $\rightarrow$ Math | 0.713 (+8.5%) | 0.7967 (+0.27%) | 0.8033 (+0.8%) |

merely memorize task-specific orders, but learns transferable scheduling principles about which positions are more informative to reveal early. The gains are especially pronounced for Code $\rightarrow$ Math in the 256-256 setting, while remaining positive across smaller block sizes.

### 6.6. Comparison with Other Order Learning Methods

Our framework learns an unmasking-order policy using a *dense* self-aware reward derived from the diffusion model's pathwise likelihood (Section 4). An alternative is to apply RLVR-style policy optimization (Guo et al., 2025) directly to the order policy using a *sparse terminal* reward, e.g., $r \in \{0, 1\}$ indicating whether the final answer is correct (Jazbec et al., 2025; Hong et al., 2025). We compare our dense self-aware reward against a $0/1$ terminal reward for learning the order policy on GSM8K under our settings.

For the terminal-reward baseline, we assign reward 1 if the generated final answer matches the ground-truth answer and 0 otherwise, and optimize the same GRPO objective. Figure 3 reports 5-shot GSM8K accuracy comparison across generation lengths and decoding regimes. Overall, our dense self-aware reward is consistently comparable and often better across settings, suggesting stability from dense, model-consistent rewards. We also report our comparison results with other concurrent order learning methods (Hong et al., 2025; Jazbec et al., 2025) in Appendix D.4.

## 7. Conclusion

We introduced **Self-Aware Scheduling (SAS)**, a plug-and-play framework for optimizing the unmasking order of pre-trained masked diffusion language models. Our theoretical analysis guarantees that optimizing our objective reduces an upper bound on the target mismatch, and achieves exactness under realizability. Combined with reinforcement learning for policy optimization, SAS improves both structured and open-ended reasoning across domains and scales—from Sudoku with a 1B model to math and code reasoning with LLaDA-8B—and remains robust across generation lengths and semi-autoregressive block decoding. Overall, SAS provides a general recipe for aligning any-order diffusion inference with probabilistic objectives, and suggests a practical path toward self-improving diffusion LLMs via iterative scheduling and model refinement. A natural future direction, in light of Remark 3.5, is to combine SAS and dependency-awareness to accelerate the method further.

## Limitations

Several limitations remain. First, SAS learns from a teacher-forced self-aware reward, i.e., the frozen denoiser's pathwise likelihood on ground-truth trajectories. This creates a potential gap between training and free-running inference. Second, our current policies are trained largely on a per-task basis, so cross-task and cross-domain transfer are promising but not yet fully characterized. Third, while SAS is plug-and-play with respect to the frozen backbone at inference time, learning the order policy still requires additional training compute, and the policy introduces a small but non-zero inference overhead. Addressing these issues through better transfer, reduced teacher-forcing mismatch, and hybrid self-aware/task-level rewards is an important direction for future work.

## Impact Statement

This paper presents work whose goal is to advance the field of Machine Learning. There are many potential societal consequences of our work, none which we feel must be specifically highlighted here.

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

# Appendix Table of Contents

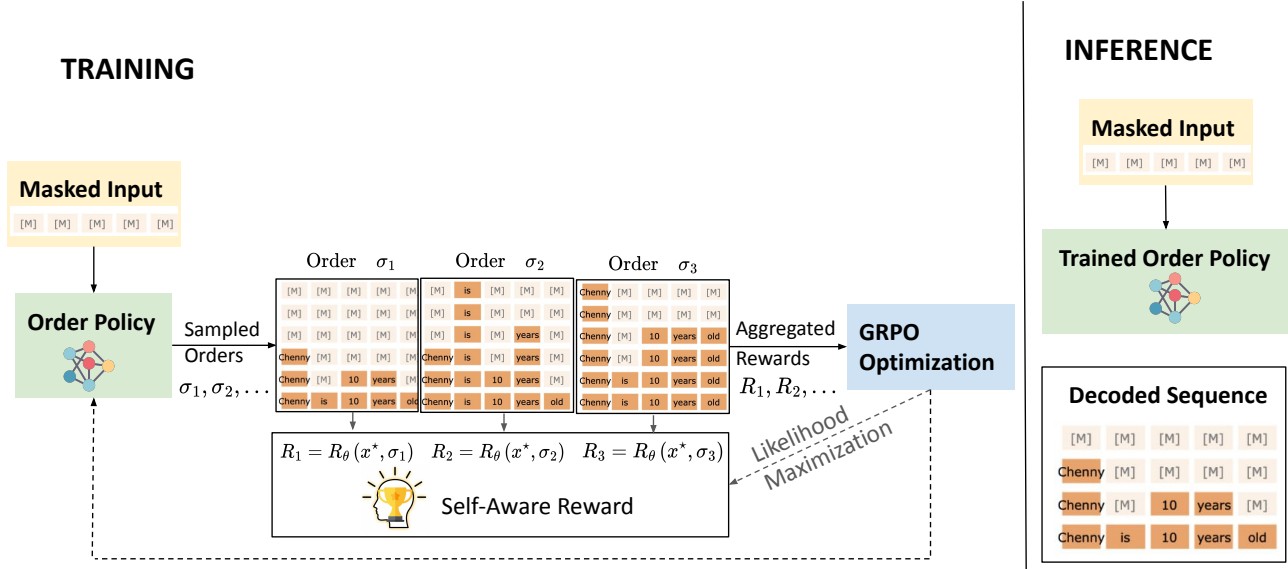

*Figure 5.* Overview of the Self-Aware Scheduling (SAS) framework. Left (Training): The order policy $v_\phi$ samples multiple unmasking permutations ($\sigma_1, \sigma_2, \dots$) for a frozen diffusion model. These orders are evaluated using our Self-Aware Reward, which measures the cumulative log-likelihood of the generation under that specific order. The policy is then optimized via GRPO to favor orders that maximize the model's own confidence. Right (Inference): The trained policy dictates the optimal unmasking sequence, guiding the diffusion model to generate high-quality outputs efficiently.

## A. Background

This section provides detailed background material supplementing the main article.

### A.1. Masked Diffusion Models

Masked diffusion language models generate discrete sequences by iteratively demasking tokens, analogous to the denoising process in continuous diffusion models. We review the modeling framework and sampling procedures.

**Notation.** Let $\mathcal{X}$ denote a finite vocabulary and $n$ the sequence length. A sequence is $x = (x_1, \dots, x_n) \in \mathcal{X}^n$. For a mask pattern $M \subseteq [n] := \{1, \dots, n\}$, define the masked sequence $x^M \in (\mathcal{X} \cup \{\texttt{[MASK]}\})^n$ by

$$(x^M)_i := \begin{cases} \texttt{[MASK]} & \text{if } i \in M, \\ x_i & \text{if } i \notin M. \end{cases} \tag{18}$$

**Model.** A masked diffusion model (or *denoiser*) parameterized by $\theta$ defines conditional distributions over masked tokens given observed context. Specifically, the model $p_\theta$ provides position-wise predictions

$$p_\theta^i(\cdot \mid x^M), \qquad i \in M, \tag{19}$$

representing the distribution over token $x_i$ conditioned on the partial observation $x^M$. Here $\theta$ denotes the learned parameters of the base model.

**Training.** Let $\pi$ denote the data distribution over $\mathcal{X}^n$ and $q(M)$ a distribution over mask patterns. The model parameters $\theta$ are trained via cross-entropy:

$$\mathcal{L}(\theta) := -\mathbb{E}_{x \sim \pi, \, M \sim q} \left[ \sum_{i \in M} \log p_\theta^i(x_i \mid x^M) \right]. \tag{20}$$

**Sampling.** Generation proceeds by iteratively unmasking positions, a process known as *decoding*. We denote by $\tilde{x}^{(t)}$ the partial sequence at decoding step $t$, where some positions contain tokens from $\mathcal{X}$ (revealed) and others contain [MASK] (not yet revealed). Starting from the fully masked sequence, at each step the algorithm selects which position(s) to unmask and samples their values from the model. The flexibility of the masked diffusion model is that it allows any order decoding in principle, and thus the unmasking order, or schedule, matters.

## A.2. Unmasking schedules

The unmasking schedule can be interpreted as an "order of thought" for generation. We introduce a learned policy $v_\phi$ parameterized by $\phi$, which takes a partial sequence as input and outputs a distribution over which position(s) to unmask next. We distinguish two decoding regimes based on how many positions are unmasked per step:

**Sequential decoding.** Unmask one position per step over $n$ steps. Let $S_t \subseteq [n]$ denote the set of revealed positions at step $t$, and $M_t = [n] \backslash S_t$ denote the set of masked positions. Starting from $\tilde{x}^{(0)} = x^{[n]}$ (fully masked) with $S_0 = \emptyset$, at each step $t = 1, \ldots, n$:

1. Select next position: $i_t \sim v_\phi(\cdot \mid \tilde{x}^{(t-1)})$ where $v_\phi(\cdot \mid \tilde{x}^{(t-1)})$ outputs a distribution over $[n] \setminus S_{t-1}$

2. Sample token: $\tilde{x}_{i_t}^{(t)} \sim p_\theta^{i_t}(\cdot \mid \tilde{x}^{(t-1)})$

3. Update: $S_t \leftarrow S_{t-1} \cup \{i_t\}$; set $\tilde{x}_j^{(t)} = \tilde{x}_j^{(t-1)}$ for all $j \neq i_t$

The complete run produces an *unmasking order* $\sigma = (i_1, \ldots, i_n)$ and final sequence $\tilde{x}^{(n)} \in \mathcal{X}^n$.

**Parallel decoding.** To accelerate generation, unmask multiple positions simultaneously. Fix a step budget $K < n$ and block sizes $(b_1, \ldots, b_K)$ with $\sum_{k=1}^K b_k = n$. Let $S_k$ denote the set of revealed positions after $k$ steps. Starting from $\tilde{x}^{(0)} = x^{[n]}$ with $S_0 = \emptyset$, at each step $k = 1, \ldots, K$:

1. Select block $B_k \subseteq [n] \setminus S_{k-1}$ of size $b_k$ (typically by sampling $b_k$ positions from $v_\phi(\cdot \mid \tilde{x}^{(k-1)})$ without replacement)

2. Sample all tokens in block: $\tilde{x}_i^{(k)} \sim p_\theta^i(\cdot \mid \tilde{x}^{(k-1)})$ for each $i \in B_k$

3. Update: $S_k \leftarrow S_{k-1} \cup B_k$; set $\tilde{x}_j^{(k)} = \tilde{x}_j^{(k-1)}$ for all $j \notin B_k$

Crucially, all positions within block $B_k$ are sampled using the *same* partial context $\tilde{x}^{(k-1)}$, enabling parallelization but ignoring correlations between positions unmasked in the same block.

**Induced distributions.** Both decoding procedures induce distributions over output sequences. To characterize these distributions, we define notations for the decoding algorithm under *teacher forcing*, where the policy $v_\phi$ and model $p_\theta$ are evaluated on partially revealed versions of a fixed target sequence $x$. This allows us to compute the probability the algorithm assigns to generating $x$ under a particular unmasking order.

For a target sequence $x$ and unmasking order $\sigma = (i_1, \ldots, i_n)$, define the *teacher-forced trajectory* $\tilde{x}^{(0)}(\sigma), \ldots, \tilde{x}^{(n)}(\sigma)$ where $\tilde{x}^{(t)}(\sigma)$ reveals positions $\{i_1, \ldots, i_t\}$ from $x$:

$$\tilde{x}^{(t)}(\sigma)_j := \begin{cases} x_j & \text{if } j \in \{i_1, \ldots, i_t\}, \\ \texttt{[MASK]} & \text{otherwise.} \end{cases} \tag{21}$$

In other words, $\tilde{x}^{(t)}(\sigma)$ is the partial sequence obtained by revealing the first $t$ positions in order $\sigma$ using their values from target $x$.

**Sequential case.** Under teacher-forcing with target $x$ and order $\sigma$, the policy (parameterized by $\phi$) assigns probability

$$v_\phi(\sigma \mid x) := \prod_{t=1}^{n} v_\phi(i_t \mid \tilde{x}^{(t-1)}(\sigma)) \tag{22}$$

to the unmasking order $\sigma$, where each factor evaluates the policy $v_\phi$ on the teacher-forced state $\tilde{x}^{(t-1)}(\sigma)$. The model (parameterized by $\theta$) assigns path likelihood

$$p_\theta(x \mid \sigma) := \prod_{t=1}^{n} p_\theta^{i_t}(x_{i_t} \mid \tilde{x}^{(t-1)}(\sigma)), \tag{23}$$

where each factor evaluates the model $p_\theta$ at position $i_t$ given teacher-forced state $\tilde{x}^{(t-1)}(\sigma)$. The sequential algorithm induces the joint distribution over sequences and orders

$$P_{\theta,\phi}^{\text{seq}}(x, \sigma) := v_\phi(\sigma \mid x) \cdot p_\theta(x \mid \sigma), \tag{24}$$

with marginal distribution over outputs

$$P_{\theta,\phi}^{\text{seq}}(x) = \sum_{\sigma \in S_n} v_\phi(\sigma \mid x) \cdot p_\theta(x \mid \sigma). \tag{25}$$

**Parallel case.** For block decomposition $\beta = (B_1, \ldots, B_K)$, define the schedule likelihood under policy parameters $\phi$:

$$v_\phi(\beta \mid x) := \prod_{k=1}^{K} v_\phi(B_k \mid \tilde{x}^{(k-1)}(\beta)), \tag{26}$$

where $\tilde{x}^{(k-1)}(\beta)$ denotes the teacher-forced state after revealing blocks $B_1, \ldots, B_{k-1}$ from target $x$. The model assigns path likelihood

$$p_\theta(x \mid \beta) := \prod_{k=1}^{K} \prod_{i \in B_k} p_\theta^i(x_i \mid \tilde{x}^{(k-1)}(\beta)). \tag{27}$$

The parallel algorithm induces the joint distribution

$$P_{\theta,\phi}^{\text{par}}(x, \beta) := v_\phi(\beta \mid x) \cdot p_\theta(x \mid \beta), \tag{28}$$

with marginal distribution

$$P_{\theta,\phi}^{\text{par}}(x) = \sum_{\beta} v_\phi(\beta \mid x) \cdot p_\theta(x \mid \beta). \tag{29}$$

### A.3. Heuristic Schedules

Heuristic schedules can be formalized as a special case of our order policy framework where the policy $v_\phi$ is **deterministic and non-trainable**. These methods define a fixed policy that greedily selects the position $i$ maximizing a predefined score computed from the model's position-wise predictive distributions $p_\theta^i(\cdot \mid \tilde{x}^{(t-1)})$. Recent works (Nie et al., 2025b; Kim et al., 2025) consider greedy schedules based on per-position uncertainty.

**Per-position uncertainty scores.** At each sequential step $t$, for a masked position $i \in M_t$, let $p(a) := p_\theta^i(a \mid \tilde{x}^{(t-1)})$ be the local probability distribution. Let $\hat{a} := \arg\max_a p(a)$ denote the most likely token. We define the following scores (higher is better):

**Confidence:** $$s_{\text{conf}}(i) := p(\hat{a}),$$

**Margin:** $$s_{\text{marg}}(i) := p(\hat{a}) - \max_{a \neq \hat{a}} p(a),$$

**Entropy:** $$s_{\text{ent}}(i) := \sum_{a \in \mathcal{X}} p(a) \log p(a).$$

---

**Algorithm 1** Unmasking-Order Policy Learning

---

**Require:** Initial order policy $v_{\phi_{\text{init}}}$; frozen diffusion model $p_\theta$; dataset $\mathcal{D}$ of pairs $(c, x^\star)$; hyperparameters $\epsilon, \beta, \mu$; group size $G$.
1: Initialize policy $\phi \leftarrow \phi_{\text{init}}$
2: **for** iteration $= 1, \ldots, I$ **do**
3:     reference policy $v_{\text{ref}} \leftarrow v_\phi$
4:     **for** step $= 1, \ldots, M$ **do**
5:         Sample minibatch $\mathcal{B} \sim \mathcal{D}$
6:         old policy $v_{\text{old}} \leftarrow v_\phi$
7:         Sample $G$ orders $\{\sigma^{(i)}\}_{i=1}^G \sim v_{\text{old}}(\cdot \mid c, x^\star)$ for each $(c, x^\star) \in \mathcal{B}$ (teacher-forced rollout)
8:         Compute self-aware returns $\{R_i\}$ using Eq. (8) (frozen $p_\theta$)
9:         Compute group-relative advantages $\{\widehat{A}_i\}$
10:         **for** GRPO update $= 1, \ldots, \mu$ **do**
11:             Update $\phi$ by maximizing the clipped GRPO objective
12:         **end for**
13:     **end for**
14: **end for**
**Ensure:** Trained order policy $v_\phi$

---

**Sequential heuristics.** At sequential step $t$, given the current state $\tilde{x}^{(t-1)}$, choose

$$i_t \in \arg \max_{i \in M_{t-1}} s(i; \tilde{x}^{(t-1)}),$$

where $s \in \{s_{\text{conf}}, s_{\text{marg}}, s_{\text{ent}}\}$.

**Parallel heuristics.** At parallel step $k$ with block size $b_k$, given the current state $\tilde{x}^{(k-1)}$, choose the block $B_k \subseteq M_{k-1}$ as the top-$b_k$ masked indices under the score:

$$B_k := \text{Top-}b_k\Big(\big\{ s(i; \tilde{x}^{(k-1)}) : i \in M_{k-1} \big\}\Big).$$

All positions in $B_k$ are then sampled from $p_\theta^i(\cdot \mid \tilde{x}^{(k-1)})$ using the same shared context.

**Discussion.** These heuristics can be viewed as deterministic policies induced by the scores and the current decoding state. While they provide a strong baseline compared to random scheduling, they remain inherently myopic: they rely exclusively on local, instantaneous uncertainty in $p_\theta^i(\cdot \mid \tilde{x}^{(t-1)})$ at the current state, and do not explicitly optimize a global objective over entire unmasking trajectories. As a result, they can prioritize "easy" positions early even when revealing a different position would better shape the future conditional landscape, leaving room for learned schedules that perform long-horizon credit assignment.

## B. Theoretical Guarantees

In this part, we provide theoretical analysis and detailed proofs of the proposed theorems and corollaries in this paper. In Appendix B.1, we first focus on the quantification of the parallel decoding errors mentioned in Remark 3.5 of the main article; in particular, we derive total variation bound in Corollary B.2. We then move to the proof of Theorem 3.2 and Corollary 3.3, in Appendices B.2 and B.3 respectively.

### B.1. Parallelization error and conditional total correlation

In this section, we provide more details on the analysis of the error caused by parallel decoding. More specifically, we will use $\text{KL}(P_{\theta,\phi}^{\text{seq}}, P_{\theta,\phi}^{\text{par}})$ to characterize the additional discrepancy introduced by parallel decoding.

We first introduce the concept of conditional total correlation.

**Conditional total correlation.** For random variables $X \in \mathcal{X}^N$ under a distribution $P$, and disjoint sets $A, B \subseteq [N]$, we define the conditional total correlation as

$$
\begin{aligned}
\mathrm{TC}_P(X_B \mid X_A) &:= \sum_{i \in B} H_P(X_i \mid X_A) \; - \; H_P(X_B \mid X_A) \\
&= \mathrm{KL}\left( P(X_B \mid X_A) \,\Big\|\, \prod_{i \in B} P(X_i \mid X_A) \right).
\end{aligned}
\tag{30}
$$

This quantity is zero if and only if the variables in $B$ are conditionally independent given $X_A$ under $P$. While parallel decoding updates a block $B_k$ using a single shared context $\tilde{x}^{(k-1)}$, which prevents within-block conditioning on newly sampled tokens. This induces an information loss whenever the true conditional dependence within $B_k$ is nontrivial.

**Theorem B.1** (Parallelization error bounded by conditional TC). *Consider a block schedule $(B_1, \ldots, B_K)$ (e.g., obtained by chunking a fine-grained order), and assume parallel updates sample each $X_i$ for $i \in B_k$ independently given the prefix $X_{B_{<k}}$. Then the additional mismatch introduced by parallelization is governed by the sum of within-block conditional total correlations under the sequential law:*

$$
\mathrm{KL}(P^{\mathrm{seq}}_{\theta,\phi}, P^{\mathrm{par}}_{\theta,\phi}) \;=\; \sum_{k=1}^{K} \mathbb{E}_{X \sim P^{\mathrm{seq}}_{\theta,\phi}} \big[ \mathrm{TC}_{P^{\mathrm{seq}}_{\theta,\phi}}(X_{B_k} \mid X_{B_{<k}}) \big].
\tag{31}
$$

*Proof.* For brevity, write $P^{\mathrm{seq}} := P^{\mathrm{seq}}_{\theta,\phi}$ and $P^{\mathrm{par}} := P^{\mathrm{par}}_{\theta,\phi}$.

**Step 1: Factorizations under the block schedule.** Fix ordered blocks $(B_1, \ldots, B_K)$ and define $B_{<k} := \bigcup_{j<k} B_j$. By the chain rule grouped by blocks, $P^{\mathrm{seq}}$ factorizes as

$$
P^{\mathrm{seq}}(x) \;=\; \prod_{k=1}^{K} P^{\mathrm{seq}}(x_{B_k} \mid x_{B_{<k}}).
\tag{32}
$$

Under the theorem assumption (within-block independent sampling given the prefix),

$$
P^{\mathrm{par}}(x_{B_k} \mid x_{B_{<k}}) \;=\; \prod_{i \in B_k} P^{\mathrm{seq}}(x_i \mid x_{B_{<k}}),
\tag{33}
$$

and therefore

$$
P^{\mathrm{par}}(x) \;=\; \prod_{k=1}^{K} \prod_{i \in B_k} P^{\mathrm{seq}}(x_i \mid x_{B_{<k}}).
\tag{34}
$$

**Step 2: Expand $\mathrm{KL}(P^{\mathrm{seq}} \| P^{\mathrm{par}})$ and identify conditional total correlation.** If $P^{\mathrm{seq}} \not\ll P^{\mathrm{par}}$, then $\mathrm{KL}(P^{\mathrm{seq}} \| P^{\mathrm{par}}) = +\infty$ and the bound is trivial. Assume henceforth $P^{\mathrm{seq}} \ll P^{\mathrm{par}}$. Then

$$
\begin{aligned}
\mathrm{KL}\big(P^{\mathrm{seq}} \| P^{\mathrm{par}}\big) &= \mathbb{E}_{X \sim P^{\mathrm{seq}}} \left[ \log \frac{P^{\mathrm{seq}}(X)}{P^{\mathrm{par}}(X)} \right] \\
&= \mathbb{E}_{X \sim P^{\mathrm{seq}}} \left[ \sum_{k=1}^{K} \log P^{\mathrm{seq}}(X_{B_k} \mid X_{B_{<k}}) - \sum_{k=1}^{K} \sum_{i \in B_k} \log P^{\mathrm{seq}}(X_i \mid X_{B_{<k}}) \right] \\
&\qquad\qquad\qquad\qquad \text{(by (32) and (34))} \\
&= \sum_{k=1}^{K} \mathbb{E}_{X \sim P^{\mathrm{seq}}} \left[ \log \frac{P^{\mathrm{seq}}(X_{B_k} \mid X_{B_{<k}})}{\prod_{i \in B_k} P^{\mathrm{seq}}(X_i \mid X_{B_{<k}})} \right].
\end{aligned}
$$

By the KL form of conditional total correlation (with respect to $P^{\mathrm{seq}}$),

$$
\mathrm{TC}_{P^{\mathrm{seq}}}(X_{B_k} \mid X_{B_{<k}}) = \mathbb{E}_{X \sim P^{\mathrm{seq}}} \left[ \log \frac{P^{\mathrm{seq}}(X_{B_k} \mid X_{B_{<k}})}{\prod_{i \in B_k} P^{\mathrm{seq}}(X_i \mid X_{B_{<k}})} \right],
$$

and hence we obtain the exact identity

$$\mathrm{KL}\big(P^{\mathrm{seq}}\|P^{\mathrm{par}}\big) \;=\; \sum_{k=1}^{K} \mathbb{E}_{X\sim P^{\mathrm{seq}}}\big[\mathrm{TC}_{P^{\mathrm{seq}}}\big(X_{B_k}\mid X_{B_{<k}}\big)\big]. \tag{35}$$

$\square$

The theorem explains why parallel decoding can incur an intrinsic error when blocks contain highly dependent variables (high conditional TC), motivating schedules that reveal strongly coupled variables in a dependency-aware manner.

**Quantification of error from data distribution**  Theorem B.1 quantifies the *parallelization gap* between the sequential and parallel output laws. We now relate this to the final deviation of parallel decoding from the data distribution $\pi$ using total variation distance.

**Corollary B.2** (Parallel decoding error via sequential error + TC). *For any $(\theta, \phi)$ and any ordered block schedule $(B_1, \ldots, B_K)$ satisfying the within-block conditional independence assumption of Theorem B.1,*

$$\mathrm{TV}\big(\pi, P^{\mathrm{par}}_{\theta,\phi}\big) \;\leq\; \sqrt{\tfrac{1}{2}\,\mathrm{KL}\big(\pi\|P^{\mathrm{seq}}_{\theta,\phi}\big)} \;+\; \sqrt{\tfrac{1}{2}\sum_{k=1}^{K}\mathbb{E}_{X\sim P^{\mathrm{seq}}_{\theta,\phi}}\Big[\mathrm{TC}_{P^{\mathrm{seq}}_{\theta,\phi}}\big(X_{B_k}\mid X_{B_{<k}}\big)\Big]}. \tag{36}$$

*Proof.* By the triangle inequality of total variation,

$$\mathrm{TV}\big(\pi, P^{\mathrm{par}}_{\theta,\phi}\big) \leq \mathrm{TV}\big(\pi, P^{\mathrm{seq}}_{\theta,\phi}\big) + \mathrm{TV}\big(P^{\mathrm{seq}}_{\theta,\phi}, P^{\mathrm{par}}_{\theta,\phi}\big).$$

Applying Pinsker's inequality to each term yields

$$\mathrm{TV}\big(\pi, P^{\mathrm{seq}}_{\theta,\phi}\big) \leq \sqrt{\tfrac{1}{2}\,\mathrm{KL}\big(\pi\|P^{\mathrm{seq}}_{\theta,\phi}\big)}, \qquad \mathrm{TV}\big(P^{\mathrm{seq}}_{\theta,\phi}, P^{\mathrm{par}}_{\theta,\phi}\big) \leq \sqrt{\tfrac{1}{2}\,\mathrm{KL}\big(P^{\mathrm{seq}}_{\theta,\phi}\|P^{\mathrm{par}}_{\theta,\phi}\big)}.$$

Finally, Theorem B.1 gives the exact identity

$$\mathrm{KL}\big(P^{\mathrm{seq}}_{\theta,\phi}\|P^{\mathrm{par}}_{\theta,\phi}\big) = \sum_{k=1}^{K}\mathbb{E}_{X\sim P^{\mathrm{seq}}_{\theta,\phi}}\Big[\mathrm{TC}_{P^{\mathrm{seq}}_{\theta,\phi}}\big(X_{B_k}\mid X_{B_{<k}}\big)\Big],$$

which proves (36). $\square$

**Discussion.**  Corollary B.2 separates the parallel decoding error into (i) a *sequential* mismatch term that can be reduced by improving the denoiser and/or learning better schedules (as in Theorem 3.2), and (ii) an *intrinsic parallelization term* controlled by within-block conditional dependence. In particular, the parallelization term vanishes only when the tokens within each block are conditionally independent given the prefix (i.e., zero conditional total correlation), explaining why overly aggressive parallel updates can degrade accuracy.

*Remark* B.3.  The first term in (36) can be viewed as the statistical learning error, while the second term corresponds to the numerical error. The design of order schedules in discrete diffusion models can impact both sources of error. In contrast, for continuous diffusion models, scalar schedules affect only the numerical error and lead to statistically equivalent models under KL divergence in path space (Chen et al., 2025; Kingma et al., 2021).

### B.2. Proof of Theorem 3.2

We then proceed to prove that our proposed objective, $\mathcal{L}_{\mathrm{SAS}}$, bounds the Kullback-Leibler divergence between the data distribution and the model-induced distribution, up to a constant term.

**Theorem 3.2** (Self-Awareness as Joint Distribution Matching). *Let $Q_\phi(x, \sigma) \coloneqq \pi(x)v_\phi(\sigma|x)$ and $P_{\theta,\phi}(x, \sigma) \coloneqq p_\theta(x|\sigma)v_\phi(\sigma|x)$ be the data-policy and model-policy joint distributions, respectively. The self-aware loss $\mathcal{L}_{\mathrm{SAS}}$ satisfies the following identity and bound:*

$$\begin{aligned} \mathrm{KL}(\pi(x)\|P^{\mathrm{seq}}_{\theta,\phi}(x)) &\;\leq\; \mathcal{L}_{\mathrm{SAS}}(\theta, \phi) - H(\pi) \\ &\;=\; \mathrm{KL}(Q_\phi(x, \sigma)\|P_{\theta,\phi}(x, \sigma)), \end{aligned} \tag{37}$$

*where $H(\pi)$ is the constant data entropy. Consequently, minimizing $\mathcal{L}_{\text{SAS}}$ is equivalent to minimizing the joint distributional mismatch, which upper bounds the generation error $\text{KL}(\pi(x)\|P^{\text{seq}}_{\theta,\phi}(x))$.*

*Proof.* Recall the induced sequential marginal

$$P^{\text{seq}}_{\theta,\phi}(x) = \sum_{\sigma \in S_n} v_\phi(\sigma \mid x)\, p_\theta(x \mid \sigma),$$

and the joint distributions defined as

$$Q_\phi(x,\sigma) = \pi(x)\, v_\phi(\sigma \mid x), \qquad P_{\theta,\phi}(x,\sigma) = v_\phi(\sigma \mid x)\, p_\theta(x \mid \sigma).$$

**Step 1: Identity** $\text{KL}(Q_\phi\|P_{\theta,\phi}) = \mathcal{L}_{\text{SAS}}(\theta,\phi) - H(\pi)$**.** By definition,

$$\begin{aligned}
\text{KL}(Q_\phi\|P_{\theta,\phi}) &= \sum_{x,\sigma} Q_\phi(x,\sigma) \log \frac{Q_\phi(x,\sigma)}{P_{\theta,\phi}(x,\sigma)} \\
&= \mathbb{E}_{x\sim\pi,\ \sigma\sim v_\phi(\cdot|x)}\left[\log \frac{\pi(x)v_\phi(\sigma \mid x)}{v_\phi(\sigma \mid x)p_\theta(x \mid \sigma)}\right] \\
&= \mathbb{E}_{x\sim\pi}[\log \pi(x)] - \mathbb{E}_{x\sim\pi,\ \sigma\sim v_\phi(\cdot|x)}[\log p_\theta(x \mid \sigma)] \\
&= -H(\pi) + \mathcal{L}_{\text{SAS}}(\theta,\phi),
\end{aligned}$$

where $H(\pi) = -\mathbb{E}_{x\sim\pi}[\log \pi(x)]$.

**Step 2: Bound** $\text{KL}(\pi\|P^{\text{seq}}_{\theta,\phi}) \leq \text{KL}(Q_\phi\|P_{\theta,\phi})$**.** Let $Q^X_\phi$ and $P^X_{\theta,\phi}$ denote the marginals of $Q_\phi$ and $P_{\theta,\phi}$ on $x$. By construction,

$$Q^X_\phi(x) = \sum_\sigma Q_\phi(x,\sigma) = \pi(x), \qquad P^X_{\theta,\phi}(x) = \sum_\sigma P_{\theta,\phi}(x,\sigma) = P^{\text{seq}}_{\theta,\phi}(x).$$

KL divergence contracts under marginalization (data processing), hence

$$\text{KL}(\pi\|P^{\text{seq}}_{\theta,\phi}) = \text{KL}(Q^X_\phi\|P^X_{\theta,\phi}) \leq \text{KL}(Q_\phi\|P_{\theta,\phi}).$$

Combining Step 1 and Step 2 yields Eq. (11), completing the proof. $\qquad\square$

### B.3. Proof of Corollary 3.3

We then move to prove the tightness of the bound derived in Theorem 3.2 can be achieved by matching a zero joint KL divergence.

**Corollary 3.3 (Joint realizability and tightness).** *If the model family is expressive enough to allow for joint realizability—i.e., there exist parameters $(\theta^\star,\phi^\star)$ such that $\text{KL}(Q_{\phi^\star}(x,\sigma)\|P_{\theta^\star,\phi^\star}(x,\sigma)) = 0$—then the marginal mismatch $\text{KL}(\pi\|P^{\text{seq}}_{\theta,\phi})$ is zero, and the self-aware lower bound of Theorem 3.1 is tight, satisfying $\mathcal{L}_{\text{SAS}}(\theta^\star,\phi^\star) = H(\pi)$.*

*Proof.* Assume $\text{KL}(Q_{\phi^\star}\|P_{\theta^\star,\phi^\star}) = 0$. Then $Q_{\phi^\star} = P_{\theta^\star,\phi^\star}$ almost everywhere, i.e., for all $(x,\sigma)$ such that $Q_{\phi^\star}(x,\sigma) > 0$,

$$\pi(x)\, v_{\phi^\star}(\sigma \mid x) = v_{\phi^\star}(\sigma \mid x)\, p_{\theta^\star}(x \mid \sigma).$$

Therefore, for any $x$ with $\pi(x) > 0$ and any $\sigma$ with $v_{\phi^\star}(\sigma \mid x) > 0$,

$$p_{\theta^\star}(x \mid \sigma) = \pi(x). \tag{38}$$

**(i) Zero marginal mismatch.** By Theorem 3.2,

$$\text{KL}(\pi\|P^{\text{seq}}_{\theta^\star,\phi^\star}) \leq \text{KL}(Q_{\phi^\star}\|P_{\theta^\star,\phi^\star}) = 0,$$

hence $\text{KL}(\pi\|P^{\text{seq}}_{\theta^\star,\phi^\star}) = 0$, i.e., $P^{\text{seq}}_{\theta^\star,\phi^\star} = \pi$.

**(ii) Tightness of Theorem 3.1.** Fix any $x$ with $\pi(x) > 0$. From Eq. (38), we know $p_{\theta^\star}(x \mid \sigma) = \pi(x)$ for all orders sampled from $v_{\phi^\star}$. Because this term is constant with respect to $\sigma$, the expectation over $\sigma$ simply returns the value itself:

$$\mathbb{E}_{\sigma \sim v_{\phi^\star}(\cdot \mid x)}[\log p_{\theta^\star}(x \mid \sigma)] = \mathbb{E}_{\sigma \sim v_{\phi^\star}(\cdot \mid x)}[\log \pi(x)] = \log \pi(x).$$

Since $P_{\theta^\star, \phi^\star}^{\mathrm{seq}}(x) = \pi(x)$ (from step i), the LHS is also $\log \pi(x)$, proving the bound is tight.

**(iii) Value of $\mathcal{L}_{\mathrm{SAS}}$.** By Theorem 3.2 and $\mathrm{KL}(Q_{\phi^\star} \| P_{\theta^\star, \phi^\star}) = 0$,

$$\mathcal{L}_{\mathrm{SAS}}(\theta^\star, \phi^\star) - H(\pi) = 0 \quad \Rightarrow \quad \mathcal{L}_{\mathrm{SAS}}(\theta^\star, \phi^\star) = H(\pi).$$

$\square$

Corollary 3.3 states that under joint realizability this lower bound is *achievable*, yielding $\mathrm{KL}(\pi \| P_{\theta, \phi}^{\mathrm{seq}}) = 0$. This provides a principled justification for our training strategy: $\mathcal{L}_{\mathrm{SAS}}$ is not only a dense, model-consistent reward for learning the unmasking order, but fundamentally connects order optimization to distribution matching. Crucially, this implies that perfect distributional alignment does not require the diffusion model to match the data distribution across *all* possible unmasking orders; rather, it suffices to identify a single optimal ordering policy along which the model's predictions align with the ground truth.

### B.4. Proof of Remark 3.4

Remark 3.4 implies that a perfect model is order-invariant, i.e. if the model class contains a parameter $\theta^\star$ whose conditionals match the true conditionals for every teacher-forced state (i.e., $p_{\theta^\star}^i(x_i \mid \tilde{x}^{(t)}) = \pi(x_i \mid \tilde{x}^{(t)})$ for all valid masks). In this regime, the upper bound in Theorem 3.2 is tight with $\mathcal{L}_{\mathrm{SAS}}(\theta^\star, \phi) = H(\pi)$ for *any* order policy $\phi$.

*Proof.* Fix any target sequence $x \in \mathcal{X}^n$ and any order $\sigma = (i_1, \ldots, i_n) \in S_n$. Let $\tilde{x}^{(t-1)}(\sigma)$ be the teacher-forced state that reveals $\{i_1, \ldots, i_{t-1}\}$ from $x$. By the chain rule of $\pi$ along the order $\sigma$,

$$\pi(x) = \prod_{t=1}^{n} \pi\big(x_{i_t} \mid \tilde{x}^{(t-1)}(\sigma)\big). \tag{39}$$

By the assumed conditional correctness of $\theta^\star$, for every teacher-forced state and every currently masked position $i_t$, we have

$$p_{\theta^\star}^{i_t}\big(x_{i_t} \mid \tilde{x}^{(t-1)}(\sigma)\big) = \pi\big(x_{i_t} \mid \tilde{x}^{(t-1)}(\sigma)\big).$$

Multiplying over $t$ and using the definition of the path likelihood,

$$p_{\theta^\star}(x \mid \sigma) = \prod_{t=1}^{n} p_{\theta^\star}^{i_t}\big(x_{i_t} \mid \tilde{x}^{(t-1)}(\sigma)\big)$$

$$= \prod_{t=1}^{n} \pi\big(x_{i_t} \mid \tilde{x}^{(t-1)}(\sigma)\big) = \pi(x),$$

where the last equality follows from (39). Hence for any order policy $v_\phi$,

$$P_{\theta^\star, \phi}^{\mathrm{seq}}(x) = \sum_{\sigma \in S_n} v_\phi(\sigma \mid x) \, p_{\theta^\star}(x \mid \sigma) = \sum_{\sigma \in S_n} v_\phi(\sigma \mid x) \, \pi(x) = \pi(x),$$

so $P_{\theta^\star, \phi}^{\mathrm{seq}} = \pi$ and therefore $\mathrm{KL}(\pi \| P_{\theta^\star, \phi}^{\mathrm{seq}}) = 0$.

**Perfect denoiser renders order irrelevant.** Since $p_{\theta^\star}(x \mid \sigma) = \pi(x)$ for all $\sigma$, we also have $P_{\theta^\star, \phi}(x, \sigma) = v_\phi(\sigma \mid x)\pi(x) = Q_\phi(x, \sigma)$ for any $\phi$. Thus $\mathrm{KL}(Q_\phi \| P_{\theta^\star, \phi}) = 0$ and by Theorem 3.2, $\mathcal{L}_{\mathrm{SAS}}(\theta^\star, \phi) = H(\pi)$ for any order policy $\phi$.

$\square$

However, since the pretrained model $p_\theta$ is inevitably imperfect, it justifies our motivation for self-aware learning: improving either $\phi$ (finding easier generation paths) or $\theta$ (improving predictions) reduces the joint mismatch, as we observe empirically in Section 6.4.

### B.5. Parallel Decoding for Unique Solution/Dirac Measure Tasks

A key advantage of diffusion language models is their ability to decode multiple tokens in parallel. While our main theory focuses on sequential unmasking, the same self-aware objective also admits a simple interpretation for parallel decoding in unique-solution tasks, such as Sudoku. In these tasks, conditioned on the prompt or constraint $c$, the target distribution is a Dirac measure:

$$\pi(\cdot \mid c) = \delta_{x^\star(c)}. \tag{40}$$

Let $\beta = (B_1, \ldots, B_K)$ denote a block order, where each $B_k$ is a set of positions decoded in parallel at step $k$. For a fixed block order $\beta$, define the teacher-forced parallel path likelihood as

$$p_\theta^{\mathrm{par}}(x^\star \mid \beta, c) = \prod_{k=1}^{K} \prod_{i \in B_k} p_\theta^i\left(x_i^\star \mid \tilde{x}^{(k-1)}(\beta; c), c\right), \tag{41}$$

where $\tilde{x}^{(k-1)}(\beta; c)$ reveals the ground-truth tokens in the earlier blocks $B_{<k}$ and masks the remaining positions.

For unique-solution tasks, each block conditional target is also a Dirac measure:

$$\pi\left(X_{B_k} \mid X_{B_{<k}} = x_{B_{<k}}^\star, c\right) = \delta_{x_{B_k}^\star} = \prod_{i \in B_k} \delta_{x_i^\star}. \tag{42}$$

Since the output space is discrete and each model conditional distribution is parameterized by a softmax, every ground-truth token has positive probability:

$$p_\theta^i\left(x_i^\star \mid \tilde{x}^{(k-1)}(\beta; c), c\right) > 0, \qquad \forall i, k. \tag{43}$$

Hence the teacher-forced parallel path likelihood is well-defined and strictly positive:

$$p_\theta^{\mathrm{par}}(x^\star \mid \beta, c) > 0. \tag{44}$$

Therefore, for any fixed block order $\beta$, the KL divergence from the Dirac target to the parallel teacher-forced model path reduces to the negative log-likelihood of the unique solution:

$$\begin{aligned}
\mathrm{KL}(\delta_{x^\star} \,\|\, p_\theta^{\mathrm{par}}(\cdot \mid \beta, c)) &= -\log p_\theta^{\mathrm{par}}(x^\star \mid \beta, c) \\
&= -\sum_{k=1}^{K} \sum_{i \in B_k} \log p_\theta^i\left(x_i^\star \mid \tilde{x}^{(k-1)}(\beta; c), c\right).
\end{aligned} \tag{45}$$

Equivalently, defining the parallel self-aware reward as

$$R_\theta^{\mathrm{par}}(x^\star, \beta) := \sum_{k=1}^{K} \sum_{i \in B_k} \log p_\theta^i\left(x_i^\star \mid \tilde{x}^{(k-1)}(\beta; c), c\right), \tag{46}$$

we obtain

$$\mathrm{KL}(\delta_{x^\star} \,\|\, p_\theta^{\mathrm{par}}(\cdot \mid \beta, c)) = -R_\theta^{\mathrm{par}}(x^\star, \beta). \tag{47}$$

Thus, for unique-solution tasks, maximizing the parallel teacher-forced self-aware reward is exactly equivalent to minimizing the parallel path KL. This provides a direct justification for applying SAS to blockwise or parallel decoding in settings where the target solution is deterministic given the condition $c$.

## C. Implementation Details

This section details the concrete implementation of the decode order policy used in our experiments. We focus exclusively on architectural choices, data flow, masking logic, and training/inference mechanics that are not explicit from the mathematical formulation in the main paper.

## C.1. Design Goals and Constraints

The decode order policy is designed to satisfy the following practical constraints:

- **Backbone-agnostic**: the policy operates on model outputs and auxiliary statistics without modifying or backpropagating through the diffusion language model.

- **Lightweight**: the policy introduces minimal additional parameters and computational overhead relative to the frozen diffusion backbone.

- **Position-selective**: at each diffusion step, the policy selects exactly one token position to reveal, subject to validity constraints.

- **Mask-aware**: the policy must never select padding positions, prompt tokens, or already revealed tokens.

These constraints motivate a per-position scoring formulation with explicit candidate masking.

## C.2. Policy Inputs

At each diffusion step, the policy operates on a partially revealed sequence of length $L$. For each position $j \in \{1, \ldots, L\}$, we construct a compact feature vector summarizing the diffusion model's current belief about that position.

Specifically, the policy input tensor has shape $[B, L, 3]$, where $B$ is the batch size, and each feature vector consists of:

1. **Maximum token probability**: the top-1 probability under the diffusion model's predicted distribution at position $j$.

2. **Token entropy**: the entropy of the predicted token distribution, capturing uncertainty.

3. **Normalized position index**: $j/L$, providing weak positional context.

4. **Current State** Current state $s_t$ with mask on future prediction positions.

These features are computed on-the-fly during diffusion rollouts and are detached from the backbone model graph.

## C.3. Policy Architecture

The decode order policy is implemented as a small Transformer encoder that maps per-position features to scalar reveal scores.

**Encoder.** We use a Transformer encoder with (i) batch-first layout, (ii) $n_{\text{layer}}$ self-attention layers, (iii) model dimension $d_{\text{model}}$ and (iv) $n_{\text{head}}$ attention heads.

Self-attention allows the policy to reason about relative confidence and uncertainty across positions rather than scoring positions independently.

**Output Head.** The encoder output at each position is projected through a linear layer to produce a scalar $\lambda_j$, yielding a score tensor $\lambda \in \mathbb{R}^{B \times L}$. No softmax is applied at this stage; normalization is deferred until candidate masking is applied.

## C.4. Candidate Masking and Valid Actions

Not all positions are valid candidates for selection at every step. We construct a binary candidate mask $M \in \{0, 1\}^{B \times L}$ indicating valid reveal positions.

A position is considered **invalid** if it satisfies any of the following:

- it corresponds to a padding token,
- it lies in the prompt region,
- it has already been revealed in a previous diffusion step.

Invalid positions are assigned $-\infty$ before normalization to ensure they receive zero probability mass. Only positions with $M_j = 1$ participate in action selection.

## C.5. Action Selection

At each diffusion step, the policy selects exactly one position per sequence. Given masked scores $\tilde{\lambda}$, we form a categorical distribution over valid positions and sample (or greedily select) an index: $j^* = \arg\max_j \tilde{\lambda}_j$.

The selected position is then revealed in the diffusion canvas, and the process repeats until all answer tokens are unmasked.

## C.6. Training-Time Supervision

During training, we supervise the policy using ground-truth reveal orders extracted from reference diffusion trajectories.

For each diffusion step:

- the candidate mask is constructed,
- logits are normalized only over valid positions,
- GRPO loss is computed against the target reveal position.

Loss is accumulated across steps and averaged over the batch. Padding positions and invalid candidates are fully excluded from loss computation.

## C.7. Inference-Time Integration

At inference time, the policy replaces heuristic scheduling strategies. The diffusion model remains unchanged; only the reveal order differs.

The policy is queried once per diffusion step, introducing negligible overhead compared to the backbone forward pass. All experiments use greedy selection for stability.

**Implementation Notes.** All components are implemented in PyTorch. Attention masks and candidate masks are represented as boolean tensors and applied consistently in both training and inference. The policy network is trained independently and can be attached to any compatible diffusion language model without retraining the backbone.

# D. Training and Evaluation Results

In this section, we provide further details on the evaluation of our unmasking order policy across the Sudoku, GSM8K, and MBPP datasets.

## D.1. Sudoku

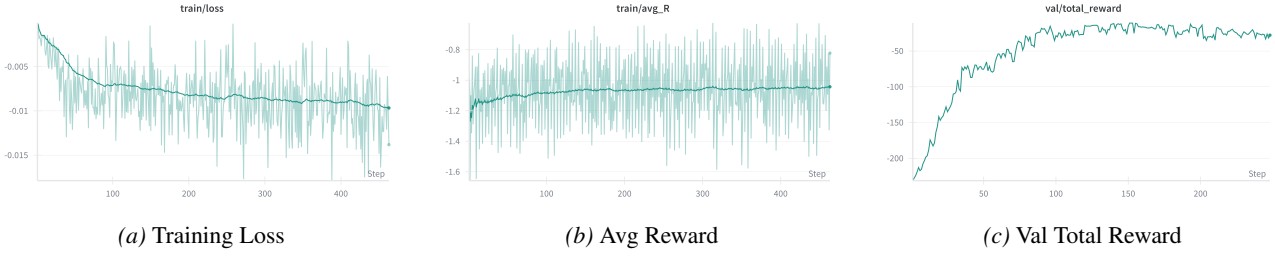

*(a)* Training Loss        *(b)* Avg Reward        *(c)* Val Total Reward

*Figure 6.* Training and Validation Metrics. (a) Training loss decreases over time. (b) Average training reward stabilizes. (c) Validation total reward converges.

**Training dynamics** We show the training and evaluation dynamics in Figure 6. It illustrates the training and validation dynamics of the proposed order policy optimization. Panel (a) depicts the training loss, which exhibits a consistent downward trend over the course of training, indicating stable optimization of the objective function. Panel (b) tracks the average training reward, showing a rapid initial increase followed by stabilization, which reflects the policy successfully learning to prioritize high-likelihood reveal orders. Finally, Panel (c) demonstrates that the total reward on the validation set follows a similar trajectory, converging to a high value; this plateauing confirms that the learned policy generalizes robustly to unseen data without significant overfitting.

**Equivalence-class Kendall's $\tau$**  We also conduct a systematic analysis of the order under different schedulings and compare them with human EXPERT order using equivalence class Kendall's $\tau$. In this paragraph, we show details on equivalence-class Kendall's $\tau$ used for order analysis in Sudoku regime (Section 6.1). Standard rank correlation metrics penalize any deviation from the reference order. However, in Sudoku, many moves are strategically equivalent. For instance, a *Naked Single* (a cell with only one valid candidate) and a *Hidden Single* (a cell is the only position in a row/col/box for a specific number) are both deterministic logical steps. If a board state offers multiple Naked Singles simultaneously, they can be filled in any order without changing the underlying strategy. We propose a variant of Kendall's $\tau$ that ignores permutations among such equivalent moves. Given a human expert order $\pi$ and a model order $\sigma$, we iterate through the expert's trajectory. Let $s_t$ be the board state after $t - 1$ expert moves. For the current expert move $\pi_t$ and any future move $\pi_j$ (where $t < j$), we check if $\pi_j$ could have been solved at step $t$ using the *same deduction logic* as $\pi_t$ (i.e., $c(\pi_j; s_t) = c(\pi_t; s_t)$). If so, the choice between $\pi_t$ and $\pi_j$ is arbitrary; we treat this pair as unordered and exclude it from the evaluation.

We calculate Kendall's $\tau$ solely on the remaining *strategy-relevant* pairs—those where the expert prioritized a simpler or more fundamental deduction class over a complex one (e.g., prioritizing a *Naked Single* over a complex *Intersection Removal*). Let $\mathcal{P}_{\text{rel}}$ be the set of relevant pairs. We compute the number of concordant pairs $C^*$ (model agrees with expert hierarchy) and discordant pairs $D^*$ (model flips hierarchy) within $\mathcal{P}_{\text{rel}}$. The metric is defined as:

$$\tau_{\text{eq}} = \frac{C^* - D^*}{C^* + D^*}.$$

A value of $\tau_{\text{eq}} = 1$ implies the model perfectly matches the expert's strategic hierarchy, even if the exact sequence differs on interchangeable steps.

**When does the model fail on Sudoku?**  We report an empirical observation on *when* sequential masked-diffusion decoding tends to make its first mistake on Sudoku. We first fine-tune the base diffusion model on the Sudoku corpus using the standard masked-diffusion pretraining objective (Section 2.1) so that it acquires basic Sudoku-solving capability. We then evaluate the resulting model by sequentially unmasking cells under a given decoding schedule (e.g., our learned order policy).

To quantify failure timing, we define the **first-step failure** as the earliest decoding step $t$ at which the generated value for the selected cell differs from the ground-truth value at that position (i.e., the first wrong fill along the rollout). Figure 7 plots the distribution of this first failure step on the evaluation set for random, confidence, margin, and learned order policy. Interestingly, for heuristic schedules, failures concentrate near the *end* of decoding (roughly the last few steps), rather than at the beginning. In contrast, we observe that a random schedule tends to fail much earlier, indicating that scheduling substantially affects not only the final accuracy but also the *error onset* during generation. Even with a trained order policy, however, the diffusion model still exhibits a nontrivial mass of late-stage failures, suggesting that the remaining unsolved cells are intrinsically harder and motivating further improvements to the diffusion head (e.g., our second-stage fine-tuning in Section 6.4).

### D.2. Math and code reasoning with LLaDA-8B

**Training Details & Computational Cost**  We trained the order policy using the Self-Aware Scheduling (SAS) framework with a group size of $G = 6$ and a batch size of $B = 3$ (total effective batch size of 18 trajectories per optimization step). The maximum sequence length was set to $L = 512$, with a learning rate of $1 \times 10^{-4}$ on 4 NVIDIA H200 GPUs. Based on the training logs, the wall-clock time per step—including group sampling, reward computation, and backpropagation—averaged approximately 3.2 minutes. The policy typically converged within 200 steps (approx. 12 GPU-hours). The computational cost of SAS training is dominated by the rollout phase, where the policy samples $G$ distinct ordering trajectories, and the reward phase, where the frozen diffusion model evaluates the pathwise log-likelihood of these trajectories via teacher-forced recomputation. The frozen denoiser's likelihood evaluation (Reward) consumes the majority of FLOPs, significantly exceeding the policy's gradient update cost.

**Evaluation**  We also report detailed evaluation result in Table 5. We use ARGMAX choice with order policy by setting the temperature 0 during evaluation. While heuristic baselines exhibit inconsistent performance across different settings, our learned order policy consistently demonstrates superior or competitive results across the entire spectrum of decoding regimes. This robustness confirms that our scheduling strategy captures fundamental reasoning structures, allowing it to transfer effectively well beyond its training configuration.

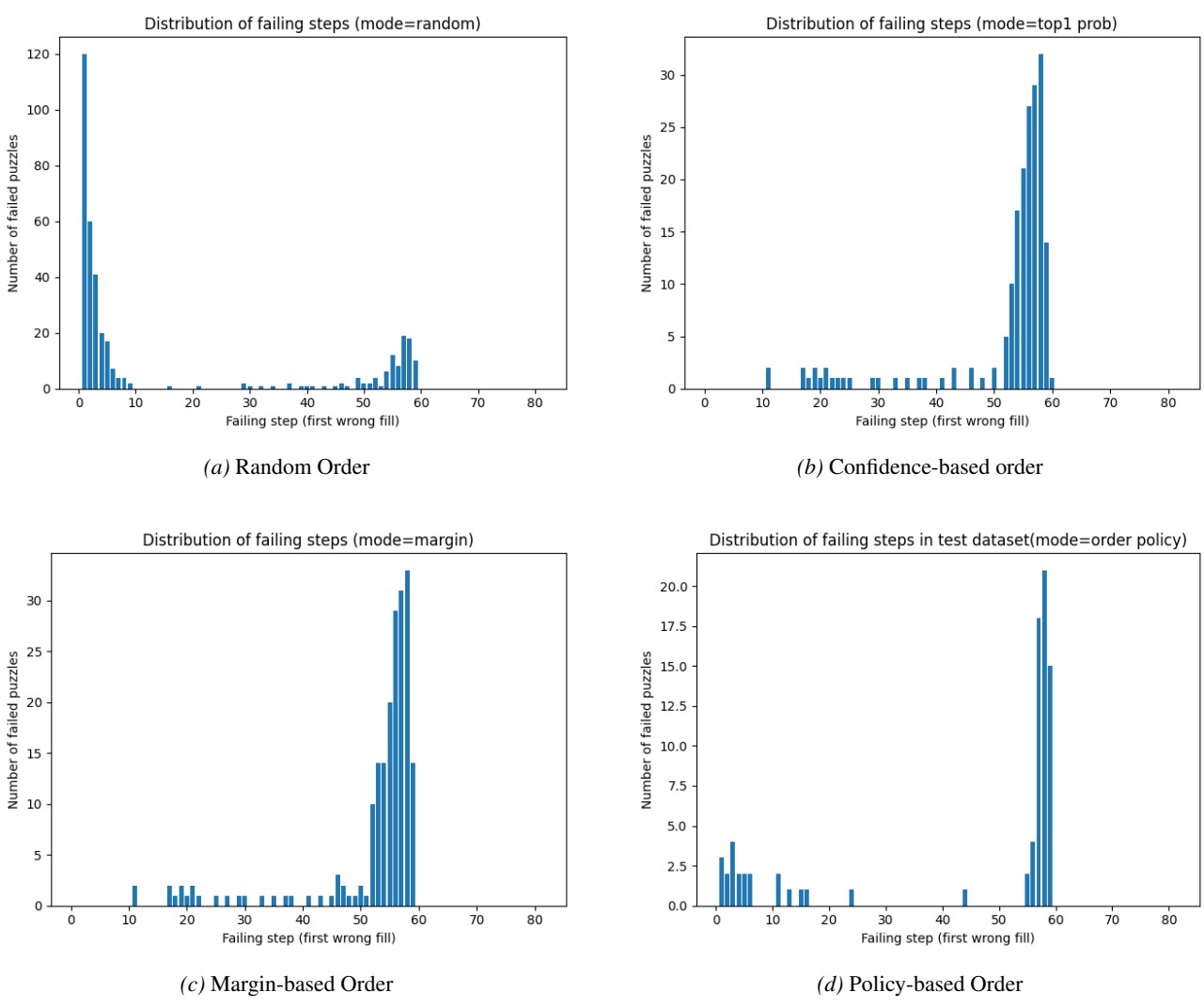

*(a)* Random Order

*(b)* Confidence-based order

*(c)* Margin-based Order

*(d)* Policy-based Order

*Figure 7.* Distribution of failing steps across different ordering strategies. (a) Random ordering fails early. (b) Confidence strategies show late-failure patterns. (c) Margin and (d) Learned policy shifts failures later, but failure still concentrates on the last 5 steps.

*Table 5.* Generalization across generation length and semi-autoregressive decoding. We report pass@1 accuracy with error bars on GSM8K (5-shot) and MBPP (3-shot). Notation: `L--B` denotes total generation length $L$ and block size $B$; `L--L` is full diffusion decoding. We vary the number of decoding blocks $K \in \{1, 4, 8\}$ across different generation lengths.

| Order | GSM8K | | | | | | | | |
|---|---|---|---|---|---|---|---|---|---|
| | 128–128 | 128–32 | 128–16 | 256–256 | 256–64 | 256–32 | 512–512 | 512–128 | 512–64 |
| Confidence (Top-1) | $0.620_{\pm0.018}$ | $0.728_{\pm0.017}$ | $0.744_{\pm0.017}$ | $0.628_{\pm0.018}$ | $0.760_{\pm0.016}$ | $0.758_{\pm0.017}$ | $0.765_{\pm0.018}$ | $0.810_{\pm0.017}$ | $0.810_{\pm0.016}$ |
| Margin | $0.608_{\pm0.019}$ | $0.742_{\pm0.017}$ | $0.740_{\pm0.018}$ | $0.640_{\pm0.018}$ | $0.794_{\pm0.016}$ | $0.796_{\pm0.017}$ | $0.628_{\pm0.020}$ | $0.800_{\pm0.017}$ | $0.815_{\pm0.016}$ |
| Entropy | $0.600_{\pm0.020}$ | $0.740_{\pm0.018}$ | $0.700_{\pm0.019}$ | $0.620_{\pm0.019}$ | $0.790_{\pm0.017}$ | $0.790_{\pm0.017}$ | $0.792_{\pm0.016}$ | $0.798_{\pm0.017}$ | $0.802_{\pm0.016}$ |
| **Order Policy (Ours)** | $\mathbf{0.660_{\pm0.017}}$ | $\mathbf{0.788_{\pm0.016}}$ | $\mathbf{0.756_{\pm0.017}}$ | $\mathbf{0.760_{\pm0.016}}$ | $\mathbf{0.810_{\pm0.016}}$ | $\mathbf{0.800_{\pm0.016}}$ | $\mathbf{0.793_{\pm0.017}}$ | $\mathbf{0.825_{\pm0.016}}$ | $\mathbf{0.827_{\pm0.016}}$ |

| Order | MBPP | | | | | | | | |
|---|---|---|---|---|---|---|---|---|---|
| | 128–128 | 128–32 | 128–16 | 256–256 | 256–64 | 256–32 | 512–512 | 512–128 | 512–64 |
| Confidence (Top-1) | $0.373_{\pm0.011}$ | $0.400_{\pm0.010}$ | $0.388_{\pm0.010}$ | $0.395_{\pm0.011}$ | $0.396_{\pm0.010}$ | $0.387_{\pm0.010}$ | $0.385_{\pm0.011}$ | $0.392_{\pm0.010}$ | $0.388_{\pm0.010}$ |
| Margin | $0.333_{\pm0.012}$ | $0.390_{\pm0.010}$ | $0.388_{\pm0.010}$ | $0.363_{\pm0.011}$ | $0.380_{\pm0.010}$ | $0.378_{\pm0.010}$ | $0.385_{\pm0.011}$ | $0.388_{\pm0.010}$ | $0.380_{\pm0.010}$ |
| Entropy | $0.323_{\pm0.012}$ | $0.388_{\pm0.010}$ | $0.396_{\pm0.010}$ | $0.388_{\pm0.011}$ | $0.385_{\pm0.010}$ | $0.378_{\pm0.010}$ | $0.390_{\pm0.011}$ | $0.390_{\pm0.010}$ | $0.393_{\pm0.010}$ |
| **Order Policy (Ours)** | $\mathbf{0.375_{\pm0.010}}$ | $\mathbf{0.405_{\pm0.009}}$ | $\mathbf{0.416_{\pm0.009}}$ | $\mathbf{0.410_{\pm0.010}}$ | $\mathbf{0.407_{\pm0.009}}$ | $\mathbf{0.400_{\pm0.009}}$ | $\mathbf{0.407_{\pm0.010}}$ | $\mathbf{0.395_{\pm0.009}}$ | $\mathbf{0.395_{\pm0.009}}$ |

*Table 6.* Inference-time overhead of SAS on GSM8K with generation length 128, averaged over 15 runs.

| Setting | Avg. decoding time |
|---|---|
| Backbone only | 338.753s |
| Backbone + SAS | 344.415s |
| Added time from SAS | +5.662s (1.67%) |

**Inference cost.** SAS introduces only a small inference-time overhead because the order policy is lightweight compared to the frozen backbone denoiser. On GSM8K with generation length 128, averaged over 15 runs, backbone-only decoding takes 338.753s, while backbone decoding with SAS takes 344.415s. Thus, SAS adds only 5.662s of decoding time, corresponding to a 1.67% total overhead. This suggests that the learned ordering mechanism improves decoding quality while preserving nearly the same inference efficiency as the original backbone.

**Direct generalization to parallel decoding.** We also evaluate whether a sequentially trained SAS policy can be directly reused for parallel top-$k$ decoding, where multiple positions are decoded at each step. Here the decoding setting is denoted by $L$-$B$-$T$, where $L$ is generation length, $B$ is block size, and $T$ is the number of decoding steps. Table 7 reports the results on GSM8K and MBPP.

These results suggest that a policy trained only for sequential decoding does not directly optimize the parallel top-$k$ decoding regime. Compared with sequential decoding, direct parallel decoding leads to performance drops of different magnitudes across tasks and block configurations. This is expected because decoding multiple tokens simultaneously introduces additional dependencies that are not present in the sequential training objective. We therefore view direct parallel policy learning as an important extension of SAS: while sequentially trained policies can be applied to parallel decoding, fully exploiting the parallel advantage of diffusion language models likely requires training the order policy directly under blockwise or parallel decoding objectives.

### D.3. Open-ended Instruction-following Generation

To evaluate whether learned scheduling also helps beyond structured reasoning tasks, we additionally train and test SAS on the open-ended instruction-following benchmark IFEval (Zhou et al., 2023). We compare the learned order policy against standard heuristic schedules under the same decoding setting, where $L$-$B$-$T$ denotes generation length, block size, and number of decoding steps. Table 8 reports results for the 256-64-256 setting. The learned order policy outperforms all heuristic baselines, suggesting that the benefit of learned scheduling is not limited to Sudoku, math, or code generation, but also extends to open-ended instruction-following generation.

### D.4. Comparison with other order learning methods

We compare SAS with prior order-learning methods (Hong et al., 2025; Jazbec et al., 2025) from two complementary perspectives. A fully controlled reimplementation of all prior methods is difficult because some codebases and training

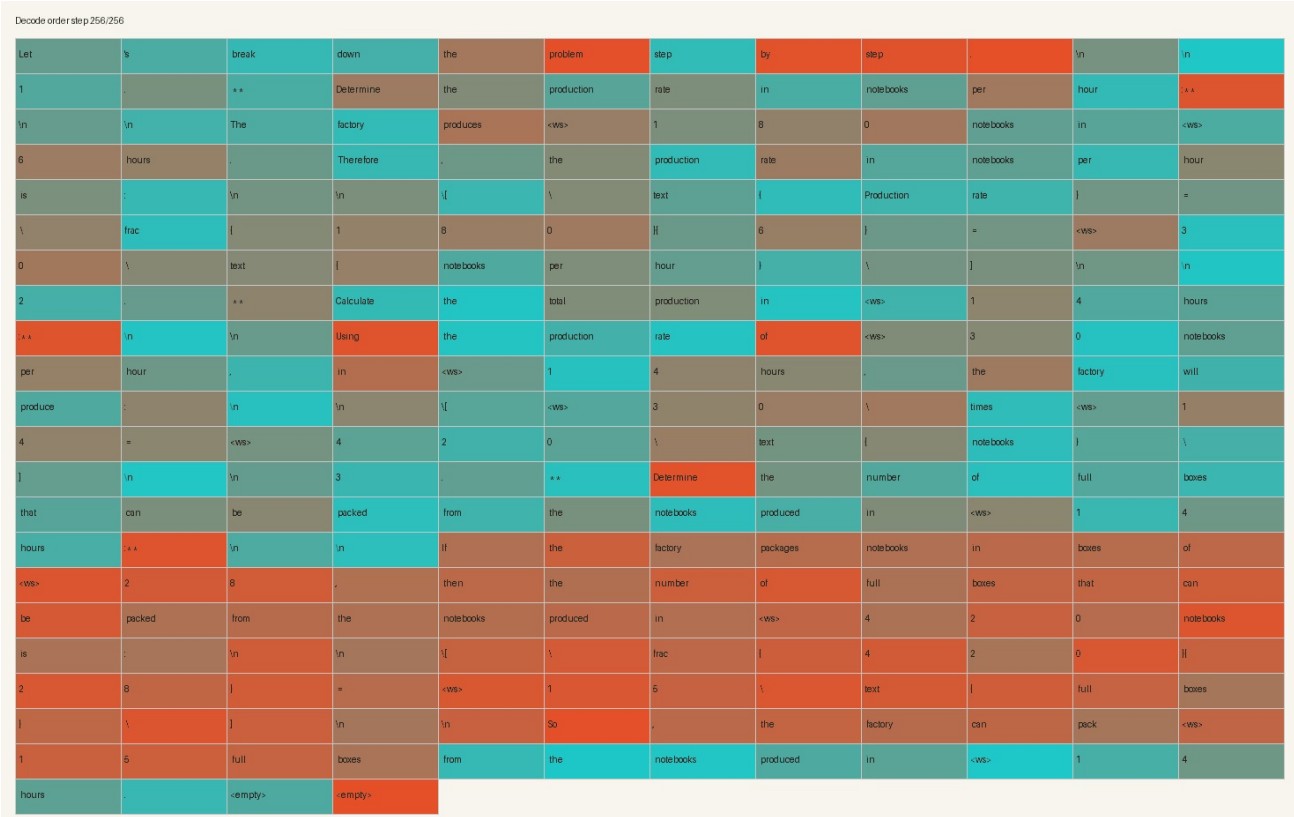

*Figure 8.* **Visualization of learned decoding order on a multi-step reasoning task.** Prompts = ["A factory produces notebooks over several days. First, determine a key parameter: on Monday, the factory produces notebooks at a constant rate and makes 180 notebooks in 6 hours. 1. Using the rate from second question, how many notebooks will the factory produce in 14 hours? 2. What is the production rate in notebooks per hour? 3. if the factory must package notebooks in boxes of 28, how many full boxes can be packed from the notebooks produced in those 14 hours?"]. The text is colored by generation step, transitioning from **blue (generated first)** to **red (generated last)**. Notably, the model prioritizes answering **Question 2** (calculating the rate) first, identifying it as a prerequisite before solving **Question 1**, which depends on that rate. This non-sequential ordering demonstrates the model's capacity for **global semantic understanding**, allowing it to plan and resolve logical dependencies effectively.

details are not publicly available, and because several prior methods rely on external verifiable terminal rewards rather than the self-aware likelihood reward used in SAS. Therefore, we avoid making claims of a definitive apples-to-apples benchmark. Instead, we provide: (i) a controlled reward ablation within our own framework, and (ii) matched-setting comparisons against reported numbers from prior work.

**Controlled terminal-reward comparison within our framework.** To isolate the key methodological difference, we replace our dense self-aware reward with a sparse terminal reward similar to those used in prior verifier-based order-learning methods, while keeping the same backbone denoiser, order-policy network, and training pipeline fixed. This comparison is reported in Section 6.5 and Figure 3. Under this controlled setup, SAS performs better overall, suggesting that the dense self-aware reward provides a more effective learning signal than sparse terminal correctness feedback for learning the unmasking order.

**Matched-setting comparison to reported results.** We also compare SAS against reported values from Hong et al. and Jazbec et al. under matched decoding settings when available. We denote each setting by $L$-$B$-$T$, where $L$ is the generation length, $B$ is the block size, and $T$ is the number of decoding steps. The results are shown in Table 9. SAS is competitive overall and stronger in most matched reported settings, especially on GSM8K. On GSM8K, SAS improves over prior reported results in two settings and matches the best reported value in the third. On MATH500, SAS outperforms Jazbec et al. in the 256-256-256 setting, while falling below the best reported value in the other two settings.

*Table 7.* Direct generalization of a sequentially trained SAS policy to parallel top-$k$ decoding. Each setting is denoted by $L$-$B$-$T$, where $L$ is generation length, $B$ is block size, and $T$ is the number of decoding steps.

| Task | 128-128-64 | 128-64-32 | 256-64-64 | 256-64-32 |
|---|---|---|---|---|
| GSM8K | 0.58 | 0.60 | 0.61 | 0.58 |
| MBPP | 0.2567 | 0.14 | 0.14 | 0.1667 |

*Table 8.* Open-ended instruction-following evaluation on IFEval. The decoding setting is 256-64-256, where $L$-$B$-$T$ denotes generation length, block size, and number of decoding steps.

| Method | IFEval Score |
|---|---|
| Confidence | $0.5067 \pm 0.0289$ |
| Margin | $0.5233 \pm 0.0289$ |
| Entropy | $0.5267 \pm 0.0288$ |
| **Order Policy (Ours)** | $\mathbf{0.5633 \pm 0.0288}$ |

**Sudoku comparison across problem scales.**     We further evaluate order learning on Sudoku, where prior work and SAS were originally tested on different problem scales. To make the comparison more informative, we train the available (Hong et al., 2025). codebase on $9 \times 9$ Sudoku and our codebase on $4 \times 4$ Sudoku, so that both methods are evaluated on both $4 \times 4$ and $9 \times 9$ settings. The results are shown in Table 10. SAS performs favorably on both scales.

**Interpretation.**     Overall, these comparisons suggest that SAS is competitive with existing order-learning methods and often improves over them in matched reported settings. However, we emphasize that the comparison is not perfectly controlled: methods may differ in backbone model, architecture, training pipeline, reward design, and task setup. We therefore view the reported comparisons as suggestive evidence rather than a definitive apples-to-apples benchmark. A fully controlled same-backbone comparison, for example on GSM8K with LLaDA, would be a valuable direction for future work.

## E. Comparison with Hidden-State–Conditioned Decode Order Policy

In addition to the feature-based order policy described in the main paper, we implement and evaluate an alternative decode-order policy that directly conditions on the internal hidden states of the frozen LLaDA diffusion model. This variant is intended to test whether richer token-level representations can further improve reveal order selection in practice.

The feature-based policy operates on lightweight summary statistics (e.g., entropy, maximum probability), which are inexpensive and model-agnostic. In contrast, hidden states encode substantially richer contextual information, including long-range dependencies and latent reasoning structure. However, they are also higher-dimensional, more entangled with the backbone model, and potentially misaligned with the decode-order decision boundary.

### E.1. Policy Inputs

At each diffusion step, we extract the last-layer hidden states from the frozen LLaDA model for all token positions. For a batch of size $B$ and sequence length $L$, this yields a tensor:

$$H \in \mathbb{R}^{B \times L \times d_{\text{hid}}}.$$

Hidden states corresponding to padding tokens are masked out and never used as candidates. No gradients are propagated into the diffusion model.

### E.2. Architecture

The hidden-state order policy is implemented as a lightweight projection network operating on $H$:

- A linear projection from $d_{\text{hid}}$ to $d_{\text{proj}}$
- A non-linear activation
- A final linear layer producing a scalar score per position

This produces per-position scores $\lambda \in \mathbb{R}^{B \times L}$, analogous to the feature-based policy. No additional self-attention layers are

*Table 9.* Matched-setting comparison with reported results from prior order-learning methods. Each setting is denoted by *L-B-T*, where *L* is generation length, *B* is block size, and *T* is the number of decoding steps. Missing entries indicate that the corresponding setting was not reported in the prior work.

| GSM8K | | | |
|---|---|---|---|
| Method | 128-32-128 | 256-32-256 | 256-256-256 |
| Hong et al. | 0.703 | – | – |
| Jazbec et al. | – | 0.800 | 0.330 |
| **SAS (ours)** | **0.7875** | **0.800** | **0.760** |
| MATH500 | | | |
| Method | 128-32-128 | 256-32-256 | 256-256-256 |
| Hong et al. | **0.284** | – | – |
| Jazbec et al. | – | **0.355** | 0.200 |
| **SAS (ours)** | 0.260 | 0.300 | **0.2467** |

*Table 10.* Cross-scale Sudoku comparison of SAS and (Hong et al., 2025). Both methods are trained and then evaluated on $4 \times 4$ and $9 \times 9$ Sudoku.

| Method | $4 \times 4$ Sudoku | $9 \times 9$ Sudoku |
|---|---|---|
| Hong et al. | 80% | 76% |
| **SAS (ours)** | **95%** | **91%** |

used, as contextualization is already encoded in the backbone hidden states.

### E.3. Candidate Masking and Action Selection

Candidate masking follows the same rules as in the feature-based policy:

- prompt tokens are excluded,
- already revealed tokens are excluded,
- padding positions are excluded.

At each step, the policy selects the highest-scoring valid position and reveals the corresponding token. All training uses the GRPO loss.

### E.4. Comparison with Feature-Based Policy

Table 11 summarizes performance on GSM8K under varying decode budgets.

*Table 11.* GSM8K accuracy under different decode budgets.

| Method | 5-shot / 128–128 | 5-shot / 128–64 | 5-shot / 128–32 |
|---|---|---|---|
| Top-1 Confidence | 0.620 | 0.711 | 0.728 |
| Margin | 0.608 | 0.752 | 0.742 |
| Entropy | 0.600 | 0.756 | 0.740 |
| Order Policy (features) | 0.660 | 0.748 | **0.788** |
| Left-to-Right | $0.754 \pm 0.019$ | $0.756 \pm 0.019$ | $0.756 \pm 0.019$ |
| Hidden-State Policy | $0.478 \pm 0.022$ | $0.700 \pm 0.021$ | $0.754 \pm 0.019$ |

**Observations.** Across settings, the hidden-state policy does not consistently outperform the simpler feature-based policy. While performance improves as the decode budget becomes more constrained, the hidden-state variant remains less robust than the feature-based order policy and often underperforms even heuristic schedules.

**Discussion.**    These results suggest that directly exposing high-dimensional backbone representations to the order policy can be counterproductive. Although hidden states contain rich semantic information, they also entangle many factors—such as token identity, syntactic structure, and semantic content—that may not be directly relevant to the decode-order decision. This can make the hidden-state policy harder to optimize and less stable. In contrast, explicitly constructed confidence and uncertainty features, such as top-token probability, margin, and entropy, provide a more direct, interpretable, and stable signal for deciding which position to reveal next, while remaining lightweight and model-agnostic. We therefore view the hidden-state variant as an exploratory design study rather than a negative result about all representation-based policies. The feature-based policy is chosen not because the design space is exhausted, but because it is the most stable, lightweight, and empirically reliable option we found.

