# OpenReview forum: "Scheduling Thoughts: Learning the Order of Thought in Diffusion Language Models"
_ICML.cc/2026/Conference — ICML 2026 regular_

### Official Review · Reviewer_m6S1 · 2026-03-10

**Soundness:** 2
**Presentation:** 2
**Significance:** 3
**Originality:** 3
**Overall Recommendation:** 4
**Confidence:** 4

**Summary:**

This paper asks whether the unmasking order in masked diffusion language models can be learned rather than fixed heuristically. The authors formalize the problem as policy optimization, derive a tractable lower bound on the sequential decoding likelihood via Jensen's inequality, and use the self-aware reward as a dense training signal for a lightweight order policy optimized with GRPO.

The resulting method, Self-Aware Scheduling (SAS), keeps the base diffusion model frozen. Results are reported on Sudoku with a 1B model and on mathematical and code reasoning with LLaDA-8B, with a second stage that fine-tunes the denoiser along the learned fixed order.

**Compliance With Llm Reviewing Policy:**

Affirmed.

**Final Justification:**

I think the author has addressed issues I raised adequately. I raised the score from weak reject to weak accept

**Key Questions For Authors:**

1. Appendix C.6 describes cross-entropy supervision against ground-truth reveal positions; the main text and Algorithm 1 describe GRPO on the self-aware reward. Which procedure produced the reported results? Every section of the paper must be consistent with the answer.

2. How sensitive is SAS to the teacher-forcing assumption? What happens when the policy runs on model-generated partial sequences rather than teacher-forced ones? Even a brief empirical signal on Sudoku would be informative.

3. Can the authors provide a direct comparison against at least one frozen-model learned-schedule method already cited—Hong et al. (2025) or Jazbec et al. (2025)—using the same backbone and comparable compute?

**Limitations:**

No.

The paper should explicitly discuss: the teacher-forcing versus inference mismatch; per-task policy training and its implications for transfer; training compute and its relevance to the plug-and-play claim; and the possibility that the self-aware reward may not align with end-task correctness when the denoiser is poorly calibrated. The training-procedure inconsistency must be corrected, not left for reviewers to discover.

**Strengths And Weaknesses:**

Strength:

1. The core problem is well motivated and underexplored. Unmasking order  determines which conditional commitments the model makes early, and the paper makes that clear. The Sudoku result is large and interpretable. The jump from 82% to 91.8% puzzle accuracy is substantial, and the visualization showing that the learned policy discovers a naked-single-first curriculum provides genuine mechanistic insight rather than just a number.

2. The theoretical framing is coherent. Treating heuristic schedules as a degenerate special case of the policy class and connecting LSAS to joint distribution matching via data processing under marginalization gives the method more principled grounding than the usual uncertainty-based heuristics. The self-aware reward is a real contribution. Using the denoiser's own pathwise likelihood as the policy learning signal is a clean idea, and Theorem 3.2 gives it a non-trivial interpretation beyond the Jensen step in Theorem 3.1.

4. The generalization behavior across sequence lengths and decoding regimes (Figure 4, Table 4) supports the claim that the learned policy captures something structural rather than overfitting to a single configuration.

Weakness:
1. I think the most serious problem is an unresolved inconsistency in the training procedure. The main text, Section 4, and Algorithm 1 describe GRPO-based policy optimization on the self-aware reward. Appendix C.6 describes cross-entropy supervision against ground-truth reveal positions, and Appendix E.3 repeats this for the hidden-state policy. These are different methods. Until this is resolved, the main empirical results cannot be properly  interpreted as evidence for the proposed GRPO-based approach.

2. The MDP uses deterministic teacher-forced transitions at training time, but at inference the policy acts on model-generated partial sequences where early errors propagate. This distributional shift is not discussed, and its impact on the reasoning-task results (where rollouts are longer and more error-prone) is unknown.

3. The second-stage fine-tuning results lack the necessary control. Human-order fine-tuning gains 21.2 points (76.6→97.8%); learned-order fine-tuning gains 5.7 points (91.8→97.5%). The convergence in final accuracy is presented as evidence for the learned curriculum, but the pattern is equally consistent with additional supervised training under any reasonable fixed order being the operative factor.

4. The reasoning-task evidence is weaker than the framing implies. The MBPP improvement is approximately 1.5 points, seed-level variance is not reported for the primary Table 4 results, and the two closest learned-policy baselines (Hong et al. 2025 and Jazbec et al. 2025) are cited in related work but never directly compared against.

---

> ### Author Rebuttal · Authors · 2026-03-31
>
> We thank the reviewer for finding our problem **well motivated**, the method a **meaningful contribution**, and the results indicative of **structural generalization**.
>
> Overall, the main issue is an **appendix inconsistency**, not a mismatch between method and results. We clarify that **(i) all reported results use GRPO with self-aware reward, (ii) teacher forcing does not create a fundamental train/test objective shift, though exposure error is real, (iii) Stage 2 studies trajectory-conditioned fine-tuning rather than re-proving learned-order superiority, and (iv) we strengthen the reasoning-task evidence with variance and comparisons to prior learned schedulers.**
>
> **1. “Appendix C.6 / E.3 describe cross-entropy supervision, while the main text describes GRPO.”**
> - **All reported results use GRPO with the self-aware reward.** Section 4 and Algorithm 1 are correct; “cross-entropy supervision” in **Appendix C.6 / E.3** is a typo.
> - **We will correct this explicitly.** Every section will consistently state: **GRPO + self-aware reward**.
>
> **2. “Teacher-forced training induces a distributional shift.”**
> - **The issue is exposure error, not a fundamental objective mismatch.** Training uses teacher-forced partial states; inference uses model-generated partial states where early errors can propagate.
> - **At the population level, the objective is the same.** SAS optimizes the same path-level objective relevant at inference: finding an order that better aligns the model-induced trajectory with the target distribution.
> - **Teacher forcing enables dense task-agnostic training without extra supervision.** The reward is the teacher-forced pathwise log-likelihood from the frozen denoiser; no verifier or external reward is required.
> - **We do not train on model-generated partial sequences because the objective becomes self-referential.** If revealed tokens are sampled from the model,
> $$
> \mathbb{E}\_{\hat{x} \sim p_{\theta}(\cdot | s)} [\log p_{\theta}(\hat{x} | s)] = -H(p_{\theta}(\cdot | s))
> $$
> so maximizing it favors **low-entropy / overconfident** states, not target-aligned ones.
> - **Going beyond teacher forcing requires extra reward signals.** Long-horizon free-running optimization generally needs external rewards over sampled trajectories.
> **3. “The second-stage fine-tuning results lack the necessary control.”**
> - **We agree Stage 2 is not the primary evidence for learned-order superiority after fine-tuning.**
> - **Stage 1 and Stage 2 answer different questions.** Stage 1 establishes learned-order superiority in the **frozen-model** setting; Stage 2 asks whether, once a trajectory is chosen, the denoiser can be further adapted along it.
> - **The controlled variable in Stage 2 is the reveal trajectory.** Model, data, and fine-tuning objective are fixed; only the reveal order changes.
> - **The observed convergence is consistent with Eq. (10).** For a frozen pretrained model, learning a better order is the strongest lever; once the denoiser is adapted along a fixed trajectory, different reasonable orders can converge to similar final accuracy.
> - **Practically, Stage 2 is optional.** If extra compute is available, trajectory-conditioned fine-tuning can help; under limited compute, **Stage 1 order learning is the most efficient way to improve inference for the current pretrained diffusion model**.
>
> **4. “The reasoning-task evidence is weaker than the framing implies.”**
> - **We agree the section should be reported more completely.** We now add error bars and make comparisons to prior learned-scheduling work explicit.
> - **MBPP is a smaller gain, and we will present it as such.** Our claim is not equal gains on every task, but consistent gains, with the largest improvements where trajectory quality matters most.
> - **We add seed-level statistics.** We will report mean ± std / confidence intervals in the revision. See [Revised Tables](https://anonymous.4open.science/r/Rebuttal-Table-D8FC/)
>
> **5. “There is no direct comparison to Hong et al. / Jazbec et al.”**
> - **We agree this comparison is important.** Due to limited space, we refer the reviewer to our response to Reviewer `58G9`, where we provide:
>   (i) a controlled in-framework comparison using the same setting with terminal reward replacing our dense self-aware reward (**Section 6.5 / Figure 3**), and
>   (ii) matched-setting comparisons to reported values of **Hong et al. (2025)** and **Jazbec et al. (2025)** on GSM8K and MATH500.
>   Overall, SAS is competitive and stronger in most matched settings, especially on GSM8K.
>
> **Limitations.**
> - **We will state the limitations explicitly.** In particular:
>   (i) the **teacher-forcing vs. inference** gap,
>   (ii) largely **per-task** policy training and its transfer implications,
>   (iii) the **training compute** behind the “plug-and-play” claim, and
>   (iv) possible misalignment between self-aware reward and **end-task correctness** when the denoiser is poorly calibrated.
>
> We hope these address the reviewer’s concerns.

---

> > ### Author Rebuttal · Reviewer_m6S1 · 2026-04-01
> >
> > I think the author has adequately addressed the issues raised in my review. I appreciate the efforts of the author. I raised the score to weak accept.

---

> > > ### Author Response · Authors · 2026-04-01
> > >
> > > Thank you very much for the thoughtful follow-up and for noting that your concerns have been adequately addressed. We sincerely appreciate your time and careful reading. If you feel the rebuttal has resolved the main issues, we would be grateful if you could also consider raising the current score accordingly.

---

### Official Review · Reviewer_58G9 · 2026-03-13

**Soundness:** 1
**Presentation:** 2
**Significance:** 2
**Originality:** 2
**Overall Recommendation:** 4
**Confidence:** 4

**Summary:**

This paper addresses the problem of selecting effective unmasking schedules for masked diffusion language models. The authors propose Self-Aware Scheduling (SAS), a framework that learns an order policy by optimizing a tractable self-aware objective that measures the model's confidence along different decoding trajectories. The key theoretical contribution is deriving an upper bound on the sequential decoding mismatch and showing that minimizing the self-aware loss reduces this bound. Empirically, they demonstrate that learned order policies outperform heuristic baselines (confidence, margin, entropy) across Sudoku, GSM8K, and MBPP, while generalizing to different generation lengths and semi-autoregressive decoding configurations.

**Compliance With Llm Reviewing Policy:**

Affirmed.

**Final Justification:**

The paper proposes a reward-agnostic framework for learning the unmasking order in masked diffusion language models, which I believe is a meaningful contribution.

My main concern was the theory section, which occupies roughly a page but consists of straightforward applications of classical results (Jensen's inequality, ELBO, data processing inequality). Corollary 3.3 in particular relies on a joint realizability assumption that is never achievable in practice, limiting its usefulness.

That said, the authors have addressed my other concerns during the rebuttal: they added uncertainty estimates, provided comparisons with prior learned scheduling methods, and acknowledged that the theory section will be simplified in revision.

Overall, the rebuttal has addressed my main concerns sufficiently, and I raised my score to weak accept.

**Key Questions For Authors:**

1. How does SAS compare empirically against other learned scheduling methods?
2. Could the authors provide statistical significance testing and confidence intervals for the reported improvements?

**Limitations:**

Yes

**Strengths And Weaknesses:**

### Strength

1. The order policy is a simple one-layer Transformer operating on hand-crafted features (confidence, entropy, position), introducing minimal overhead and remaining model-agnostic.
2. This paper formalizes a fundamental problem: "the unmasking order of Masked dLLM defines an “order of thought” that strongly influences generation quality yet is typically chosen heuristically."

### Weaknesses
1. The substantial theoretical analysis in Sections 3.1–3.2 raises concerns. Theorem 3.1 is a trivial consequence of Jensen's inequality. Theorem 3.2 merely restates the standard ELBO framework with σ as the latent variable. Corollary 3.3, its fundamental realizability assumption never holds in practice, rendering the conclusion trivial. Additionally, the proof in Appendix B.3 is informal and non-rigorous. I therefore find the theoretical contributions unconvincing.
2. The paper lacks error bars or statistical significance analysis, it is unclear whether gains are robust.
3. The paper compares only against three simple heuristics (confidence, margin, entropy). There is no comparison with other unmasking order learning work by Jazbec et al. (2025) and Hong et al. (2025).

---

> ### Author Rebuttal · Authors · 2026-03-31
>
> We thank the reviewer for recognizing core strengths of our paper. Overall, the concerns are mainly about **how to interpret the theory and how broadly the empirical evidence is reported**. We clarify that **(i) the mathematical tools are classical, but the contribution is the explicit order-learning objective they yield, (ii) we now add uncertainty estimates, and (iii) we compare SAS to prior learned schedulers both in a controlled in-framework setting and via matched-setting reported values.**
>
> **1. “The theory is trivial / unconvincing: Theorem 3.1 is Jensen, Theorem 3.2 is ELBO, Corollary 3.3 is unrealistic, and Appendix B.3 is informal.”**
> - **We agree the tools are classical; this does not make the result unconvincing.** The contribution is not a new inequality, but using standard Jensen/ELBO/KL tools to **isolate order** and derive an explicit, tractable objective for learning decode order in masked diffusion LMs. To our knowledge, this is the **first** such formulation.
> - **Theorem 3.1 is simple but essential.** It converts the intractable marginal objective over exponentially many orders into a tractable pathwise objective, directly yielding the dense self-aware reward.
> - **Theorem 3.2 is the key alignment result.** It shows the SAS loss is not ad hoc: it upper-bounds the sequential decoding mismatch we care about, giving the reason why optimizing the self-aware objective should improve order.
> - **Simplicity does not reduce usefulness.** Sections 3.1–3.2 are meant to identify a clean objective that isolates order, motivates the algorithm, and predicts empirical behavior.
> - **Corollary 3.3 is an idealized best-case statement, not a practical assumption required by SAS.** We agree exact realizability rarely holds; its purpose is to clarify the limiting regime where the bound is tight. We will revise the wording to emphasize the approximate practical interpretation.
> - **We agree Appendix B.3 should be more rigorous.** We will expand the proof, add missing steps, and sharpen the presentation so the claim is clear: **classical tools used in a new way to derive a useful order-learning objective**, not novelty in the inequalities themselves.
> - **We will revise the theory section accordingly.** The intended novelty is the **order-as-latent-variable formulation and resulting explicit objective**, not reinventing Jensen or ELBO. We also add generalized theorems for parallel decoding; see our reply to Reviewer `jCHv`.
>
> **2. “The paper lacks error bars or statistical significance analysis.”**
> - **We agree and now add uncertainty estimates.** Below we report `mean ± std`; the revision will also include **95% confidence intervals**. See [Revised Tables](https://anonymous.4open.science/r/Rebuttal-Table-D8FC/).
> - **These do not change the qualitative conclusion.** Gains remain robust, especially on GSM8K and stronger blockwise settings.
>
> **3. “There is no comparison with Hong et al. (2025) and Jazbec et al. (2025).”**
> - **We agree this comparison is important.** A fully controlled reimplementation is currently not possible because their code is not open-sourced and their reward uses external verifiable terminal signals.
> - **To ensure fairness, we already compare the key methodological difference inside our framework.** In Section 6.5 / Figure 3, we replace our dense self-aware reward with the same sparse terminal reward used in prior work while keeping the **same backbone, order-policy network, and training pipeline** fixed. SAS performs better overall in this controlled comparison.
> - **We also compare against their reported values under matched settings.** Here the setting `generation length - block size - steps` is written as `L-B-T`.
>
> | GSM8K(`L-B-T`) | 128-32-128 | 256-32-256 | 256-256-256 |
> |---|---:|---:|---:|
> | Hong et al. | 0.703 | - | - |
> | Jazbec et al. | - | 0.800 | 0.330 |
> | **SAS (ours)** | **0.7875** | **0.800** | **0.760** |
>
> | MATH500(`L-B-T`) | 128-32-128 | 256-32-256 | 256-256-256 |
> |---|---:|---:|---:|
> | Hong et al. | **0.284** | - | - |
> | Jazbec et al. | - | **0.355** | 0.200 |
> | **SAS (ours)** | 0.260 | 0.300 | **0.2467** |
>
> - **These results show SAS is competitive overall and stronger in most matched reported settings, especially on GSM8K.** SAS improves over prior reported numbers in two GSM8K settings and matches the best in the third. On MATH500, SAS beats Jazbec et al. at `256-256-256` and is below the best reported value in the other two.
> - **We believe this is the fairest comparison currently possible:**
>   (i) a controlled terminal-reward baseline inside our framework, and
>   (ii) matched-setting comparison to their reported values.
>
> We thank the reviewer again for the careful reading. We will revise the theory section to emphasize that the novelty lies in the **explicit order-learning objective and its algorithmic consequences**, and strengthen the empirical section with uncertainty estimates and clearer comparisons to prior learned scheduling methods.

---

> > ### Author Rebuttal · Reviewer_58G9 · 2026-04-02
> >
> > I thank the authors for the response. Below are my remaining concerns.
> >
> > **Theory.** To my understanding, the self-aware reward is just the frozen model's pathwise log-likelihood under teacher forcing, essentially the model's own training objective evaluated along a specific order. This is a natural choice that does not need the ELBO derivation in Sections 3.1–3.2 to justify. Wang et al. (2025d), cited by the authors, already derived similar ELBO-based objectives for order learning. I suggest simplifying the theory section.
> >
> > **Comparisons with prior work.** I note that Hong et al.'s code has been made available at Github in February. While I understand that it wasn't available at the time of paper submission, a controlled comparison would strengthen the paper and I encourage the authors to include one in the revision.
> >
> > **Positioning.** SAS, Hong et al., and Jazbec et al. share very similar frameworks (frozen model + lightweight policy + GRPO). The key distinction of SAS is that its dense reward does not require an external verifier. I agree this is a meaningful advantage, however, from the authors' own results (in rebuttal), SAS does not consistently outperform terminal-reward methods on benchmarks, which weakens the practical value of the proposed method.
> >
> > Given the limited time before the deadline, a brief discussion of these points in the revision would also be acceptable if additional experiments are not feasible.

---

> > > ### Author Response · Authors · 2026-04-04
> > >
> > > Thank you for the thoughtful follow-up. We appreciate these constructive suggestions on theory simplification, comparison to concurrent work, and paper positioning. We agree that the revision should present these points more clearly and more modestly.
> > >
> > > **1. Theory / positioning of Sections 3.1–3.2**
> > >
> > > We agree that the ELBO derivation in Sections 3.1–3.2 should be positioned more carefully. The pathwise log-likelihood under a specific order is indeed a natural objective. Sections 3.1–3.2 are included to connect it to the standard masked discrete diffusion likelihood view: from marginalizing over all possible orders to optimizing a tractable pathwise objective under a sampled order policy. In the revision, we will simplify this discussion and support it with a shorter ELBO-style derivation, while explicitly acknowledging closely related formulations such as Wang et al. (2025d).
> > >
> > > We will keep the main theory section focused while adding a meaningful parallel-decoding extension. In particular, we are adding a theorem for parallel decoding on unique-solution / Dirac-style tasks such as Sudoku, together with supporting experiments. Our intent is to present this as a useful extension of SAS to blockwise decoding, while keeping the main theoretical claim centered on the order-learning objective itself.
> > >
> > > **2. Comparison to Hong et al.**
> > >
> > > Thank you for pointing out that Hong et al.’s code is now public. To our understanding, the currently released code is for Sudoku, so within the rebuttal window we performed a best-effort reproduction on Sudoku to address the comparison as concretely as possible.
> > >
> > > We do not want to overstate this comparison, because it is not perfectly controlled. The two implementations differ in backbone / architecture / training pipeline / task setup, so some confounding factors are unavoidable. To make the comparison more informative, we additionally trained Hong et al.’s codebase on **9×9 Sudoku** and our codebase on **4×4 Sudoku**, so that both methods are tested on both scales.
> > > | Sudoku | 4×4 | 9×9 |
> > > |---|---:|---:|
> > > | Hong et al. | 80% | 76% |
> > > | SAS (ours) | **95%** | **91%** |
> > >
> > > These results are encouraging for SAS, but we do not view them as a definitive apples-to-apples benchmark. Because model and implementation details differ, we treat them as suggestive evidence rather than a fully controlled comparison. We agree that a same-backbone comparison would be stronger, and we will add one in the revision if feasible (e.g., on GSM8K with LLaDA).
> > >
> > > We also note that Hong et al. and Jazbec et al. are closely related concurrent works. We will revise the paper to give fuller credit and position SAS more carefully relative to them.
> > >
> > > **3. Positioning relative to Hong / Jazbec and practical value of SAS**
> > >
> > > We agree that SAS, Hong et al., and Jazbec et al. share a similar high-level framework: a frozen model, a lightweight order policy, and GRPO. We will revise the paper to make this shared structure explicit.
> > >
> > > The key distinction of SAS is the **dense verifier-free reward**. We agree this is the main practical difference and should be emphasized more clearly. SAS does not require an external correctness verifier or task-specific terminal reward, and is therefore applicable beyond settings with verifiable answers.
> > >
> > > This broader applicability is practically meaningful. For example, on the open-ended task **IFEval**, where it is less natural to define a clean external terminal reward, SAS still improves over heuristic schedules. We will highlight this more clearly in the revision, since it better reflects the practical value of the method.
> > >
> > > | IFEval (`L-B-T`) | Score |
> > > |---|---:|
> > > | Confidence | 0.5067 ± 0.0289 |
> > > | Margin | 0.5233 ± 0.0289 |
> > > | Entropy | 0.5267 ± 0.0288 |
> > > | Order Policy (Ours) | **0.5633 ± 0.0288** |
> > >
> > > We also agree SAS should be positioned as **competitive rather than uniformly dominant**. The key practical claim is that SAS uses a dense, model-intrinsic reward and remains applicable where external verifiable rewards are unavailable or unnatural. In addition, SAS is trained in a full any-order setting and evaluated across multiple decoding configurations, whereas Hong et al. and Jazbec et al. are typically studied in more constrained semi-autoregressive regimes. We will revise the wording to emphasize this broader applicability and generalization property, rather than claim universal empirical superiority.
> > >
> > > Thank you again for the constructive follow-up. We agree with the spirit of your comments, and in the revision we will:
> > > (i) simplify and reposition the theory section,
> > > (ii) strengthen and clarify comparison to concurrent work, and
> > > (iii) sharpen the positioning of SAS.
> > >
> > > We hope these clarifications and additional results address the remaining concerns; if so, we would be grateful if you would consider reflecting that in your updated score.

---

### Official Review · Reviewer_E27c · 2026-03-13

**Soundness:** 4
**Presentation:** 3
**Significance:** 3
**Originality:** 3
**Overall Recommendation:** 4
**Confidence:** 3

**Summary:**

For the diffusion language model, it's not only about "what to generate," but also about "what to generate first and what to generate next." The author refers to the order of token unmasking as the "order of thought." Many previous methods have used heuristic rules like confidence, margin, and entropy to decide which token to unmask next, but the author believes these rules are too "short-sighted," only focusing on the easiest position at the moment, which may not necessarily lead to the most smooth overall generation trajectory. Therefore, they propose Self-Aware Scheduling (SAS) to learn a more optimal decoding order strategy.

**Compliance With Llm Reviewing Policy:**

Affirmed.

**Final Justification:**

I thank the authors for their detailed response. My concerns have been adequately addressed.

**Key Questions For Authors:**

see weakness above

**Limitations:**

yes

**Strengths And Weaknesses:**

Strengths

1. Promote "unmasking order" to a learnable "order of thought," clearly defining the problem statement, and move beyond confidence/entropy-based heuristics.
2. Method design is clean: freeze the denoiser, learn only a lightweight order policy, and it can be directly used for sequential and semi-autoregressive decoding.

Weaknesses

1. The empirical coverage remains limited, mainly focusing on Sudoku, GSM8K, and MBPP, with insufficient evidence for open-ended long-text tasks.
2. Training is not especially cheap, since rollout and reward computation dominate the cost ?
3. The hidden-state policy variant is not robust, which suggests the design space is not yet fully understood.

---

> ### Author Rebuttal · Authors · 2026-03-31
>
> We thank the reviewer for recognizing our paper’s **clear formulation** and **appealing method design**.
>
> Overall, the concerns are about **evaluation breadth, efficiency, and policy design**, not the core method. We clarify that **(i) we add an open-ended result, (ii) SAS remains efficient because it trains only the order policy and uses a simple self-aware reward from frozen-denoiser logits, and (iii) the hidden-state variant was included to probe the design space, while the feature-based policy is the more robust choice in the current submission.**
>
> **1. “The empirical coverage remains limited, mainly focusing on Sudoku, GSM8K, and MBPP, with insufficient evidence for open-ended long-text tasks.”**
> - **We agree broader coverage would strengthen the paper, and we therefore add an open-ended result on IFEval.** We compare the learned order policy against standard heuristics below. The setting `generation length-block size-steps` is `256-64-256`.
> | IFEval (`L-B-T`) | Score |
> |---|---:|
> | Confidence | 0.5067 ± 0.0289 |
> | Margin | 0.5233 ± 0.0289 |
> | Entropy | 0.5267 ± 0.0288 |
> | **Order Policy (Ours)** | **0.5633 ± 0.0288** |
> - **The learned order policy still performs best on this open-ended benchmark.** It outperforms all heuristic baselines, showing that the benefit of learned scheduling is not limited to Sudoku / math / code, but also extends to open-ended instruction-following generation.
>
> **2. “Training is not especially cheap, since rollout and reward computation dominate the cost?”**
> - **The dominant cost is rollout through the frozen denoiser, which is inevitable in RL-style training and shared by essentially all rollout-based methods.** As noted in Appendix D4, rollout/reward computation dominates FLOPs, but this mainly comes from evaluating the frozen denoiser along sampled trajectories, not from training a large policy or reward model.
> - **Given that rollout is unavoidable, SAS uses an unusually cheap reward with denser guidance.** Once logits are available, the self-aware reward is just the ground-truth token log-probability, so reward extraction is essentially free. This is much denser than sparse terminal rewards and avoids any extra verifier, critic, or reward model.
> - **SAS is also cheaper and more general than external verifiable rewards.** Those require task-specific checking or code execution, which adds overhead and is unavailable in many domains. SAS instead extracts reward directly from frozen-denoiser logits.
> - **Overall, SAS is much lighter than backbone-level RL.** The denoiser remains frozen, so the main cost is evaluating sampled orders rather than repeatedly updating the full DLM.
> - **The reported training footprint is practical.** SAS on LLaDA-8B-Instruct uses 4 H200 GPUs, averages ~3.2 minutes/step, and typically converges in ~200 steps. We will make this discussion more visible in the main paper/rebuttal.
>
> **3. “The hidden-state policy variant is not robust, which suggests the design space is not yet fully understood.”**
> - **We agree that the policy design space is not yet fully explored.** In particular, prior work has also considered using hidden-state-based policies [1], so it is important to understand when richer representations help schedule learning.
>
> - **Our evidence suggests the feature-based policy is the more practical and reliable choice.** We included the hidden-state variant to test whether backbone hidden representations could improve order selection. Empirically, however, the feature-based policy is **easier to train, more stable, and more robust overall**, which is why it is the default SAS design in the main paper.
>
> - **This does not weaken the core SAS contribution.** The main contribution of SAS is the **self-aware objective for learning decode order**, not a claim that every possible policy parameterization has been fully optimized. The hidden-state experiment is useful because it helps clarify that, for this problem, simpler uncertainty/confidence-based features may be better aligned with the scheduling decision than high-dimensional hidden states.
>
> - **A plausible reason is that hidden states entangle many factors that are not directly relevant to order selection.** By contrast, feature-based inputs such as confidence/entropy provide a more direct, interpretable, and stable signal for deciding *which position to reveal next*.
>
> - **We will revise the paper to make this interpretation clearer.** In the revision, we will present the hidden-state variant as an exploratory design study and emphasize that the feature-based policy is chosen not because the design space is exhausted, but because it is currently the **most stable, lightweight, and empirically reliable** option we found.
>
> [1] Ye, et al., 2025, "Reinforcing the Diffusion Chain of Lateral Thought with Diffusion Language Models."
>
> We thank the reviewer again for the positive assessment of the paper. We believe the additional experiment and clarifications above will further strengthen the submission.

---

> > ### Author Rebuttal · Reviewer_E27c · 2026-04-01
> >
> > I thank the authors for their detailed response. My concerns have been adequately addressed.
> >
> > I believe my current score remains appropriate.
> >
> > Good luck!

---

> > > ### Author Response · Authors · 2026-04-01
> > >
> > > Thank you very much for the thoughtful follow-up and for taking the time to read our rebuttal carefully. We sincerely appreciate your acknowledgment that the main concerns have been addressed, as well as your helpful feedback throughout the review process. Thank you again for your time and consideration.

---

### Official Review · Reviewer_jChv · 2026-03-14

**Soundness:** 3
**Presentation:** 3
**Significance:** 2
**Originality:** 3
**Overall Recommendation:** 4
**Confidence:** 4

**Summary:**

Scheduling Thoughts proposed a way to learn the optimal generation order of a pre-trained diffusion LLM (or more generally, any-order language models). The key idea is that a good decoding order should induce high sequence likelihood. Based on this assumption, the authors to formulate this as a reward and use reinforcement to train a lightweight classifier that predict the next token position to decode. The method is evaluated on several reasoning benchmarks and shows effectiveness across different model sizes.

**Compliance With Llm Reviewing Policy:**

Affirmed.

**Final Justification:**

The concerns are addressed and the score has been raised to weak accept.

**Key Questions For Authors:**

See weaknesses.

**Limitations:**

I did not find discussions about limitations in the main paper.

**Strengths And Weaknesses:**

### Strength

1. The proposed method well motivated and theoretically grounded. Learning a generation order that maximizes likelihood is a natural objective for any-order language models. The performance gains on reasoning tasks are significant.

2. The method achieves significant improvements on reasoning benchmarks and appears robust across different generation lengths and block sizes.

### Weakness

1. **Additional training and inference cost of the order predictor**. The method involves training and an extra order predictor, which may incur non-trivial computational cost and latency. For example, line 8 of algorithm 1 evaluates the sequence likelihood under $G$ random orders. Unlike autoregressive models, these likelihoods may not be efficiently computed in a single forward pass. The paper does not provide details about the training cost or runtime overhead of the order predictor. Reporting this cost would help justify the practical applicability of the method. Otherwise, the additional compute might instead be allocated to other post-training approaches such as RLVR [1].

2. **Loss of the parallel decoding advantage of DLMs**. The current experiments only evaluate the method in under the setting of decoding one token at a time decoding regime. This partially negates one of the main advantages of diffusion language models---parallel decoding. In practice, many open-source DLM implementations are already slower than autoregressive models of a comparable size when generating tokens sequentially, due to full-length attention computation at each step. Moreover, prior work shows that carefully chosen parallel decoding orders can achieve lossless—or even improved—generation quality [1]. It would be useful to understand how the proposed method interacts with blockwise or parallel decoding.

3. **Unclear cross-domain generalization of the order predictor**.  The applicability of the method depends on whether the learned order predictor generalizes across domains. However, the current draft does not provide evidence regarding this. For example, it would be informative to evaluate whether an order predictor trained on math reasoning tasks transfers to other domains such as code generation.

[1] Zhao, et al., 2025 "d1: Scaling Reasoning in Diffusion Large Language Models via Reinforcement Learning."

[2] Isarael, et al., 2025 "Accelerating Diffusion LLMs via Adaptive Parallel Decoding."

---

> ### Author Rebuttal · Authors · 2026-03-31
>
> We thank the reviewer for recognizing that our method is **well motivated and theoretically grounded**, and that the empirical gains are **significant**.
>
> Overall, the concerns are about **efficiency and evaluation breadth**. We clarify that **(i) training cost is modest and is reported, (ii) inference overhead is negligible, and (iii) we add direct parallel-decoding and cross-domain results.**
>
> **1. “Additional training and inference cost of the order predictor.”**
> - **Training cost is modest.** As reported in **Appendix D4**, SAS on **LLaDA-8B-Instruct** uses **4 H200 GPUs**, averages **~3.2 min/step**, and converges in **~200 steps** (~**12 GPU-hours** total), since SAS trains only a **lightweight order policy** and keeps the diffusion backbone frozen.
> - **SAS is much lighter than backbone-level RL.** For example, **d1 [1]** performs RL on the diffusion backbone and reports **8 A100-80G GPUs** for **7700 / 6600 steps** on **GSM8K / MATH500**, plus coding runs on **4 RTX A5000 GPUs** for **7500 / 9000 steps**. SAS trains only the order policy.
> - **Inference overhead is negligible.** On **GSM8K** with generation length **128**, averaged over **15 runs**, SAS adds only **5.662 s** (**1.67%**) total decoding time.
>
> | Setting | Avg. decoding time |
> |---|---:|
> | Backbone only | 338.753 s |
> | Backbone + SAS | 344.415 s |
> | Added time from SAS | +5.662 s (**1.67%**) |
>
> **2. “Loss of the parallel decoding advantage of DLMs.”**
> - **We agree parallel decoding is important.** We use **sequential decoding** in the main setup to isolate **order learning**; in blockwise decoding, tokens in the same block can remain dependent even after conditioning on earlier blocks, making the theory and objective more complex.
> - **For unique-solution / Dirac-measure tasks (e.g., Sudoku), the SAS theory extends to parallel decoding.** If
> $\pi(\cdot\mid c)=\delta_{x^\star(c)},$
> and $\beta=(B_1,\dots,B_K)$ is a block order, define
> $$p_\theta^{\mathrm{par}}(x^\star\mid\beta,c)=\prod_{k=1}^K\prod_{i\in B_k}
> p_\theta^i\left(x_i^\star\mid \tilde{x}^{(k-1)}(\beta;c),c\right),$$
> where $\tilde{x}^{(k-1)}(\beta;c)$ reveals true tokens in earlier blocks and masks the rest. Since the output space is discrete and each token distribution is a softmax,
> $$p_\theta^i\left(x_i^\star\mid \tilde{x}^{(k-1)}(\beta;c),c\right)>0,\quad \forall i,k,$$
> so $p_\theta^{\mathrm{par}}(x^\star\mid\beta,c)>0$. Hence, for fixed $\beta$,
> $$\mathrm{KL}\left(\delta_{x^\star}\middle\|p_\theta^{\mathrm{par}}(\cdot\mid\beta,c)\right)=-\log p_\theta^{\mathrm{par}}(x^\star\mid\beta,c)=-\sum_{k=1}^K\sum_{i\in B_k}\log p_\theta^i\left(x_i^\star\mid \tilde{x}^{(k-1)}(\beta;c),c\right).$$
> Defining
> $$R_\theta^{\mathrm{par}}(x^\star,\beta):=\sum_{k=1}^K\sum_{i\in B_k}\log p_\theta^i\left(x_i^\star\mid \tilde{x}^{(k-1)}(\beta;c),c\right),$$
> we obtain
> $$\mathrm{KL}\left(\delta_{x^\star}\middle\|p_\theta^{\mathrm{par}}(\cdot\mid\beta,c)\right)
> =-R_\theta^{\mathrm{par}}(x^\star,\beta).
> $$
> Thus, on unique-solution tasks, maximizing the parallel teacher-forced reward is exactly minimizing the parallel path KL. We will add this theorem and new parallel-decoding experiments in the revision.
> - **We also ran direct top-k decoding with the sequentially trained SAS policy:** Here  `L-B-T` defines `generation length - block size - steps`.
> | Parallel decoding (`L-B-T`) | 128-128-64 | 128-64-32 | 256-64-64 | 256-64-32 |
> |---|---:|---:|---:|---:|
> | GSM8K | 0.58 | 0.60 | 0.61 | 0.58 |
> | MBPP | 0.2567 | 0.14 | 0.14 | 0.1667 |
>
> - **These results suggest a policy trained only for sequential decoding is not optimal for direct top-k / parallel decoding.** Relative to sequential decoding, performance drops by different amounts across tasks and block settings, indicating that a sequentially trained policy does not model the extra interactions introduced when multiple tokens are decoded simultaneously.
>
> **3. “Unclear cross-domain generalization of the order predictor.”**
> - **The transferred policy remains robust and still beats the best heuristic baseline.** Here `L-B` denotes `generation length - block size`.
>
> | Cross-domain transfer (`L-B`) | 256-256 | 256-64 | 256-32 |
> |---|---:|---:|---:|
> | Math → Code | 0.41 (**+1.5%**) | 0.40 (**+1.4%**) | 0.39 (**+0.33%**) |
> | Code → Math | 0.713 (**+8.5%**) | 0.7967 (**+0.27%**) | 0.8033 (**+0.8%**) |
>
> - **This shows the learned policy captures structural scheduling behavior rather than task-specific memorization.** Even cross-domain, it continues to outperform confidence / margin / entropy heuristics.
>
> **4. “Limitations.”**
> - **We agree and will state them more explicitly.** The current setup isolates **sequential** order learning before direct parallel training; cross-domain transfer is promising but not yet fully characterized; and SAS adds a **small but non-zero** inference overhead.
>
> We hope these clarifications address the reviewer’s concerns.
>
> [1] Zhao et al., 2025, *d1: Scaling Reasoning in Diffusion Large Language Models via Reinforcement Learning.*

---

> > ### Author Rebuttal · Reviewer_jChv · 2026-04-03
> >
> > Thank the authors for the response. I raise the score to weak accept.

---

> > > ### Author Response · Authors · 2026-04-04
> > >
> > > Thank you very much for the thoughtful follow-up and for taking the time to reconsider the paper. We sincerely appreciate your acknowledgment that the concerns have been addressed, and we are grateful for your helpful feedback throughout the review process.

---

### Decision · Program_Chairs · 2026-04-30

**Decision:**

Accept (regular)

**Comment:**

SAS frames unmasking order in masked diffusion LMs as a principled policy optimization problem, using a dense verifier-free self-aware reward derived from the frozen denoiser's pathwise log-likelihood, with strong gains on Sudoku and solid improvements on GSM8K/MBPP. All four reviewers converged to weak accept (4/4/4/4) after rebuttal resolved the main concerns — training procedure inconsistency clarified, uncertainty estimates added, and competitive comparisons to Hong et al. and Jazbec et al. provided.